# SAISA: Towards Multimodal Large Language Models with Both Training and Inference Efficiency

## Abstract

Multimodal Large Language Models (MLLMs) mainly fall into two architectures, each involving a trade-off between training and inference efficiency: embedding space alignment (*e.g.* LLaVA) is inefficient during inference, while cross-attention space alignment (*e.g.* Flamingo) is inefficient in training. In this paper, we compare these two architectures and identify key factors for building efficient MLLMs. A primary difference between them lies in how attention is applied to visual tokens, particularly in their interactions with each other. To investigate whether attention among visual tokens is necessary, we propose a new self-attention mechanism, NAAViT (**N**o **A**ttention **A**mong **Vi**sual **T**okens), which eliminates this type of attention. Our pilot experiment on LLaVA-1.5 shows that attention among visual tokens is highly redundant. Based on these insights, we introduce SAISA (**S**elf-**A**ttention **I**nput **S**pace **A**lignment), a novel architecture that enhances both training and inference efficiency. SAISA directly aligns visual features with the input spaces of NAAViT self-attention blocks, reducing computational overhead in both self-attention blocks and feed-forward networks (FFNs). Compared with the LLaVA-1.5 architecture, SAISA reduces the inference FLOPs by 66% and the training budget by 26%, while achieving superior performance in terms of accuracy. Comprehensive ablation studies further validate the effectiveness of SAISA across various LLMs and visual encoders. The code and models will be publicly available.

## 1 Introduction

Multimodal Large Language Models (MLLMs) (OpenAI, 2023; Liu et al., 2024a; Bai et al., 2023b; Chen et al., 2023a; Dai et al., 2023) have shown impressive capabilities in understanding and processing visual information. They typically build on pre-trained Large Language Models (LLMs) (Achiam et al., 2023; Chiang et al., 2023; Bai et al., 2023a; Touvron et al., 2023b) and align visual features with the LLMs. There are two primary architectures for aligning visual and text modalities: embedding space alignment and cross-attention space alignment. Embedding space alignment, *e.g.* LLaVA (Liu et al., 2023; 2024a), introduces a projector to align visual features with the LLM embedding space and feeds the visual and text tokens into the LLM. Cross-attention space alignment, *e.g.* Flamingo (Alayrac et al., 2022), inserts cross-attention blocks and aligns visual features with the attention spaces of these blocks.

However, despite the promising performance of these MLLMs, they involve a trade-off between training and inference efficiency. On the one hand, MLLMs with embedding space alignment exhibit training efficiency, since they introduce only a small number of new parameters to the pre-trained LLMs. For example, LLaVA-1.5-7B is trained in 108 GPU hours from Vicuna-7B (Chiang et al., 2023). However, this architecture significantly increases the number of input tokens, and the computational costs of self-attention grow quadratically with the number of tokens, leading to inefficiency during inference. On the other hand, MLLMs with cross-attention space alignment achieve inference efficiency, since they do not require unrolling visual tokens, but they are inefficient during training for introducing a large number of new parameters to the pre-trained LLM. In this paper, we take a step towards building MLLMs with efficiency during both training and inference.

In this paper, we first perform a thorough analysis of these two architectures, identifying key factors for building MLLMs with both training and inference efficiency. To optimize training efficiency, the key factor is minimizing the number of new parameters and employing modules in the pre-trained LLMs for interaction between visual and text modalities. For improving inference efficiency, the main focus is reducing the computational costs associated with visual tokens, particularly in attention blocks and feed-forward networks (FFNs). Based on the analysis of these factors, we introduce NAAViT (**N**o **A**ttention **A**mong **Vi**sual **T**okens), as shown in Figure 6, a self-attention mechanism which eliminates attention among visual tokens to enhance efficiency. Since attention among visual tokens contributes significantly to the quadratically growing computational cost in self-attention blocks, we investigate whether this type of attention is truly essential for MLLMs. Our pilot experiment on LLaVA-1.5 demonstrates that NAAViT outperforms the vanilla self-attention, indicating that attention among visual tokens is highly redundant.

Based on the findings above, we introduce SAISA (**S**elf-**A**ttention **I**nput **S**pace **A**lignment), an architecture for MLLMs with efficiency during both training and inference. As illustrated in Figure 2(c), SAISA employs NAAViT blocks for multimodal interaction and directly aligns the visual features with the input spaces of these blocks. SAISA not only reduces the computational overhead of self-attention blocks, but also significantly lowers the computational costs of FFNs by eliminating the need to apply FFNs to visual tokens. We validate its effectiveness on various LLMs and visual encoders. As shown in Figure 1, SAISA outperforms LLaVA-1.5 architecture in terms of performance, training efficiency, and inference efficiency. Using Vicuna-7B-v1.5 (Chiang et al., 2023) as LLM and CLIP-ViT-L/14-336 (Radford et al., 2021) as visual encoder, SAISA reduces training budget by 26% and inference FLOPs by 66%, while delivering superior performance. Moreover, SAISA is compatible with techniques to reduce the visual token number, *e.g.* resampler (Jaegle et al., 2021).

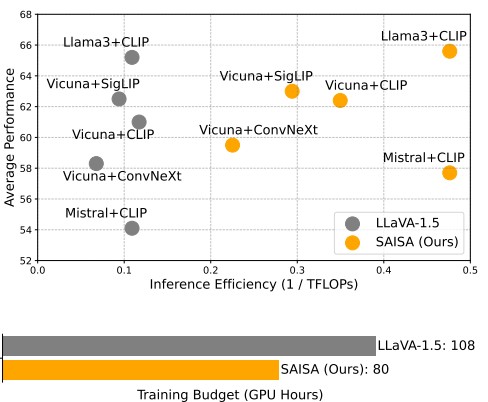

Figure 1: Top: Performance *vs.* inference efficiency based on various LLMs and visual encoders where Average Performance means an average of benchmark scores and inference efficiency is the inverse of inference TFLOPs. Bottom: Training budget comparison where we report the training GPU hours, using Vicuna-7B as LLM and CLIP as visual encoder.

Our contributions are three-fold: (1) Based on our analysis of current MLLM architectures, we propose NAAViT to enhance efficiency of MLLMs, revealing the redundancy of self-attention in MLLMs. (2) We introduce SAISA, an architecture for building MLLMs with both training and inference efficiency by eliminating the computational costs of attention among visual tokens and FFNs on visual tokens. (3) We validate the effectiveness of SAISA across various LLM backbones and visual encoders.

## 2 ANALYZING CURRENT MLLM ARCHITECTURES

In this section, we analyze the two most common architectures to align visual features with the language model, and summarize key factors for building efficient MLLMs.

**Embedding Space Alignment.** As illustrated in Figure 2(a), models with this architecture introduce a projector to align visual tokens with the text token embedding space. They concatenate visual tokens and text tokens, and then feed them into the LLM. Notable models with this architecture include LLaVA-1.5 (Liu et al., 2024a), Qwen series (Bai et al., 2023b; 2025), BLIP series (Li et al., 2023c; Dai et al., 2023), and InternVL series (Zhu et al., 2025). These models introduce only a small number of new parameters to the pre-trained LLMs, allowing training MLLMs from the LLMs with minimal budget.

However, the concatenated token sequence leads to inference inefficiency. When the number of visual and text tokens is $v$ and $t$ respectively, the computational complexity of self-attention is

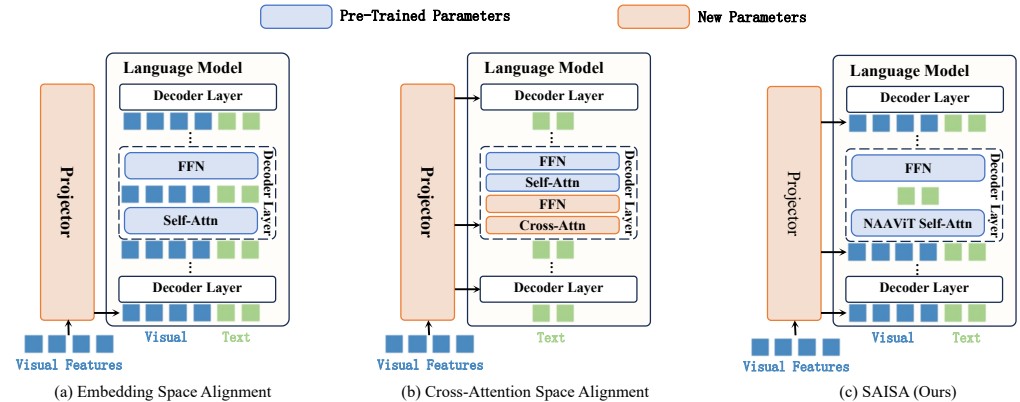

Figure 2: Overview of SAISA and mainstream architectures to align visual features with language models. (a) Aligning visual features with the embedding space of language models is inefficient during inference, *e.g.* LLaVA. (b) Aligning visual features with the attention spaces of new cross-attention blocks is inefficient during training, *e.g.* Flamingo. (c) SAISA aligns visual features with the self-attention input spaces of language models, achieving efficiency during both training and inference.

$\mathcal{O}((v + t)^2)$. It consists of three components, $\mathcal{O}(v^2)$ for attention among visual tokens, $\mathcal{O}(vt)$ for the interaction between text and visual tokens, and $\mathcal{O}(t^2)$ for attention among text tokens. Typically, MLLMs use hundreds or even thousands of visual tokens. For example, LLaVA-1.5 uses 576 visual tokens for a single image, and Sphinx (Lin et al., 2023) uses 2,890. In contrast, the number of text tokens is much smaller in most tasks. The average numbers of text tokens in MMMU (Yue et al., 2024), POPE (Li et al., 2023d) and ScienceQA IMG (Lu et al., 2022) are 142, 68, and 210, respectively. As a result, the attention among visual tokens dominates the computational overhead. Moreover, since the FFNs have a large hidden layer dimension, applying FFNs to visual tokens also brings substantial computational costs.

**Cross-Attention Space Alignment.** As illustrated in Figure 2(b), models with this architecture insert cross-attention blocks and FFNs into the language model, and align visual features with the attention spaces of these cross-attention blocks. Notable models with this architecture include Flamingo (Alayrac et al., 2022), OpenFlamingo (Awadalla et al., 2023) and Otter (Li et al., 2023a). In these models, the attention operation consists of only two components, the interaction between text and visual modalities with complexity $\mathcal{O}(vt)$ in the cross-attention blocks, and the attention among text tokens with complexity $\mathcal{O}(t^2)$ in the self-attention blocks. Compared with embedding space alignment, there is no attention among visual features in the language model. By not executing attention among visual tokens and not applying FFNs to visual tokens, these models are more efficient during inference.

However, the inserted cross-attention blocks and FFNs introduce a large number of new parameters to the pre-trained language model. As a result, training an MLLM with this architecture requires a large amount of data. For example, OpenFlamingo-9B adds 2 billion parameters to Llama-7B (Touvron et al., 2023a), and requires training data with 180M samples. In terms of model capabilities, previous work (Dai et al., 2024) finds that models utilizing this architecture perform worse than those using embedding space alignment when trained on the same data.

Based on the analyses above, we summarize the key factors for building efficient MLLMs as follows: (1) Reducing the number of new parameters and employing pre-trained modules for multimodal interaction lead to efficiency during training. (2) Reducing computations related to visual tokens, including those in attention blocks and FFNs, leads to efficiency during inference.

## 3 NO ATTENTION AMONG VISUAL TOKENS

In this section, we propose NAAViT (No Attention Among Visual Tokens) self-attention and perform a pilot experiment to investigate whether attention among visual tokens is necessary for MLLMs.

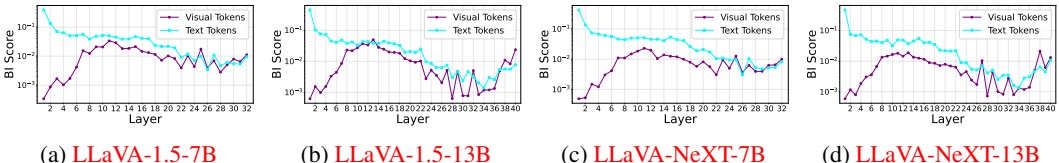

(a) LLaVA-1.5-7B      (b) LLaVA-1.5-13B      (c) LLaVA-NeXT-7B      (d) LLaVA-NeXT-13B

Figure 3: Self-attention blocks' average $BI$ scores on different types of tokens.

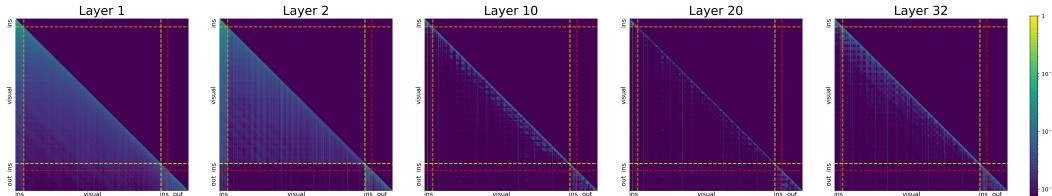

Figure 4: Attention heatmaps at different layers for LLaVA-1.5-7B.

## 3.1 PRELIMINARY

**Vanilla Self-Attention.** In a self-attention block, the input visual-text token sequence is formed as $X = [V, T] \in \mathbb{R}^{(v+t) \times h}$, where $[\cdot, \cdot]$ denotes concatenation along the sequence dimension. To derive the query, key, and value representations, three linear layers are applied to obtain $X_q$, $X_k$, and $X_v$, respectively:

$$X_q = [V,T]W_Q, X_k = [V,T]W_K, X_v = [V,T]W_V \tag{1}$$

Then, the attention operation is executed as:

$$\text{Attention}(X) = \text{softmax}\left(\frac{X_q X_k^T}{\sqrt{d}}\right) X_v \in \mathbb{R}^{(v+t) \times h} \tag{2}$$

Typically, a causal attention mask is applied and queries can only attend to keys preceding them in the sequence. The outputs are multiplied by another linear layer $W_O$ to update the hidden states through a residual connection:

$$\text{SA}(X) = \text{Attention}(X)W_O + X. \tag{3}$$

## 3.2 REDUNDANCY ANALYSES

As analyzed in Section 2, embedding space alignment achieves superior performance and training efficiency. However, it is inefficient during inference, attributable to the self-attention among visual tokens. In contrast, cross-attention space alignment does not perform attention among visual tokens in the language model, leading to inference efficiency. A question arises here: **Is attention among visual tokens redundant?** To answer this question, we present the following analyses.

**Attention Blocks in Shallow Layers Have Low Transformations on Visual Tokens** Typically, the redundancy of a block on a token is measured by cosine similarity (He et al., 2024; Men et al., 2025). Specifically, when the $i$-th token passes through a block $B$, and the input and output hidden state features are $X$ and $Y$ respectively, the block's influence ($BI$) on the token is computed as

$$BI_{B,i} = 1 - CosineSim(X, Y), \tag{4}$$

where $CosineSim(X, Y)$ denotes the cosine similarity between feature $X$ and feature $Y$. A lower $BI$ score implies that the block has a lower transformation on the token, suggesting greater redundancy. We measure each self-attention block's average $BI$ score on different types of tokens. As shown in Figure 3, self-attention blocks in the shallow layers have minimal transformations on visual tokens, lower than those on text tokens. This finding suggests that applying self-attention blocks to visual tokens in the shallow layers is redundant.

**Visual Tokens Receive Minimal Attention from MLLM's Output Tokens in Deep Layers** As illustrated in Figure 4, we visualize the attention distribution across different layers. In deep layers,

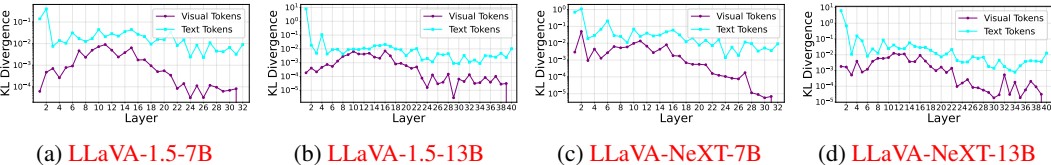

(a) LLaVA-1.5-7B     (b) LLaVA-1.5-13B     (c) LLaVA-NeXT-7B     (d) LLaVA-NeXT-13B

Figure 5: Self-attention blocks' contributions to MLLMs' output, measured by KL ivergence.

visual tokens receive low attention from output tokens, with more than 80% of the attention allocated to text tokens. This observation is consistent with previous work (Chen et al., 2024a).

**Considering the two findings above**, we can conclude that: (1) **In the shallow layers**, self-attention blocks have minimal transformations on visual tokens, indicating significant redundancy. (2) **In deeper layers**, since visual tokens receive low attention, the transformations on these tokens make little **contribution** to the model's output. To be more specific, although some self-attention blocks in deeper layers apply relatively stronger transformations on visual tokens, these tokens receive low attention in the subsequent layers, rendering such transformations ineffective for the model's output.

To quantify such **contribution** more directly, we utilize KL divergence as a metric, inspired by previous work (Lin et al., 2024). Specifically, for an attention block $B$, we eliminate $B$'s transformations on different types of tokens, respectively, *i.e.* keeping the hidden states of investigated tokens unchanged in $B$, and compute the KL divergence between the resulting model's output logits and those of the original model. A lower KL divergence indicates that removing such transformations in the investigated block has minimal impact on the model's output. As shown in Figure 5, the attention blocks' transformations on visual tokens consistently contribute less than those on text tokens. This observation inspires us to introduce NAAViT in the following subsection, which eliminates attention blocks' transformations on visual tokens.

We attribute such redundancy to: (1) MLLMs apply causal masking to both text and visual tokens. While texts are inherently sequential, images are two-dimensional. Consequently, imposing such attention to visual tokens introduces unnecessary noise. (2) The visual encoder already effectively captures the interactions among visual tokens.

## 3.3 NAAViT

We propose NAAViT (No Attention Among Visual Tokens), which eliminates attention among visual tokens. We illustrate the architecture of NAAViT in Figure 6 (Top). Specifically, for the visual-text token sequence $X = [V, T] \in \mathbb{R}^{(v+t) \times h}$, queries $X_q$, keys $X_k$ and values $X_v$ are obtained as:

$$X_q = TW_Q, X_k = [V, T]W_K, X_v = [V, T]W_V \quad (5)$$

The attention operation is executed as Equation 2, but with NAAViT attention mask, where the queries can attend to visual tokens and text tokens preceding them.

When the number of visual and text tokens is $v$ and $t$ respectively, the vanilla self-attention exhibits a computational complexity of $\mathcal{O}((v+t)^2)$ for the attention operation in Equation 2. By eliminating the attention among visual tokens, NAAViT reduces the complexity to $\mathcal{O}(t(v+t))$. Furthermore, since NAAViT reduces the query length from $v+t$ to $t$, it also significantly reduces the computational costs associated with linear layers $W_Q$ and $W_O$.

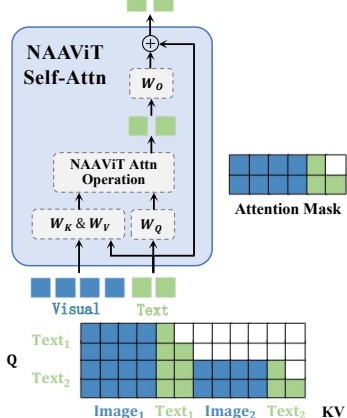

Figure 6: Top: NAAViT self-attention block. Bottom: Attention mask for interleaved image-text.

## 3.4 PILOT EXPERIMENT

We train a model under the same configurations as LLaVA-1.5-7B, but replace the vanilla self-attention with NAAViT.

In Table 1, we compare NAAViT and vanilla self-attention on multiple MLLM benchmarks, including MMMU (Yue et al., 2024), MM-Bench (Liu et al., 2024b), MMBench-CN (Liu et al., 2024b), POPE (Li et al., 2023d), Sci-

| Attention | MMMU VAL | MMBench EN | MMBench CN | POPE | ScienceQA IMG | OK-VQA |
|---|---|---|---|---|---|---|
| Vanilla | 35.7 | 64.3 | 58.3 | 86.8 | 66.8 | 53.4 |
| NAAViT | 36.0 | 64.9 | 58.4 | 86.9 | 68.7 | 56.0 |

Table 1: Effects of attention mechanisms.

enceQA IMG (Lu et al., 2022) and OKVQA (Marino et al., 2019). Among them, MMMU contains interleaved text-image inputs. For these inputs, the NAAViT attention mask is illustrated in Figure 6 (Bottom). Despite eliminating the attention among visual tokens, the model employing NAAViT outperforms the one using the vanilla self-attention. Given that NAAViT substantially reduces computational overhead, it offers a favorable balance between performance and efficiency.

In conclusion, attention among visual tokens is highly redundant for building MLLMs. In the following section, we introduce SAISA (Self-Attention Input Space Alignment) for efficient MLLMs based on NAAViT.

## 4 SAISA

### 4.1 ARCHITECTURE

As mentioned earlier, besides the attention operation, another factor that contributes to inference inefficiency is applying FFNs to visual tokens. Based on NAAViT, which eliminates the attention among visual tokens, we propose SAISA (Self-Attention Input Space Alignment), an architecture for further enhancing MLLM efficiency. In SAISA, we also eliminate FFNs' computations on visual tokens.

We illustrate the SAISA architecture in Figure 2(c). SAISA contains a visual encoder to extract visual features, a projector, and an LLM. Each layer of the LLM consists of a self-attention block and an FFN. We utilize NAAViT in the self-attention blocks. The purpose of the projector is to directly align the visual features with the input spaces of different self-attention blocks in the LLM.

Specifically, we assume $n$ is the number of layers in the LLM, $h$ is the hidden size of the LLM, $d$ is the dimension of visual features, $v$ is the number of visual tokens, and $t$ is the number of text tokens. For an input image $I$, we first employ the visual encoder VE to extract visual features:

$$Z = \text{VE}(I) \in \mathbb{R}^{v \times d} \tag{6}$$

Then, we use the projector P to directly align the visual features with the input spaces of different NAAViT self-attention blocks:

$$V = \text{P}(Z) \in \mathbb{R}^{n \times v \times h} \tag{7}$$

For the $i$-th NAAViT self-attention block $\text{NAAViT}_i$, we input the aligned visual features $V_i \in \mathbb{R}^{v \times h}$ and hidden states of the text tokens $T_i \in \mathbb{R}^{t \times h}$:

$$H_i = \text{NAAViT}_i(V_i, T_i) \in \mathbb{R}^{t \times h}, i = 1, 2, 3, \ldots, n \tag{8}$$

Notably, the NAAViT self-attention block only outputs the text tokens' hidden states $H_i$ for the subsequent FFN, and we apply the FFN to update the text tokens:

$$T_{i+1} = \text{FFN}_i(H_i) \in \mathbb{R}^{t \times h}, i = 1, 2, 3, \ldots, n. \tag{9}$$

SAISA utilizes $T_{n+1}$ for the next-token prediction.

### 4.2 PROJECTOR

Since each LLM layer operates in distinct attention spaces, the projector must flexibly align the visual features with each of the spaces. For simplicity, we employ distinct two-layer MLPs for each layer of the LLM. When the LLM has $n$ layers, the projector contains $n$ MLPs. Following LLaVA-1.5 (Liu et al., 2024a), we set the intermediate size of each MLP to be the same as the LLM hidden size.

Specifically, for the $i$-th layer of the LLM,

$$V_i = \text{MLP}_i(Z) = \varphi(ZW_{i,1})W_{i,2}, i = 1, 2, 3, \ldots, n \tag{10}$$

where $Z \in \mathbb{R}^{v \times d}$ is the visual features from the visual encoder, $\varphi$ is the activation function like GELU (Hendrycks & Gimpel, 2016), $W_{i,1} \in \mathbb{R}^{d \times h}$ and $W_{i,2} \in \mathbb{R}^{h \times h}$ are the weight matrices of the two fully connected layers.

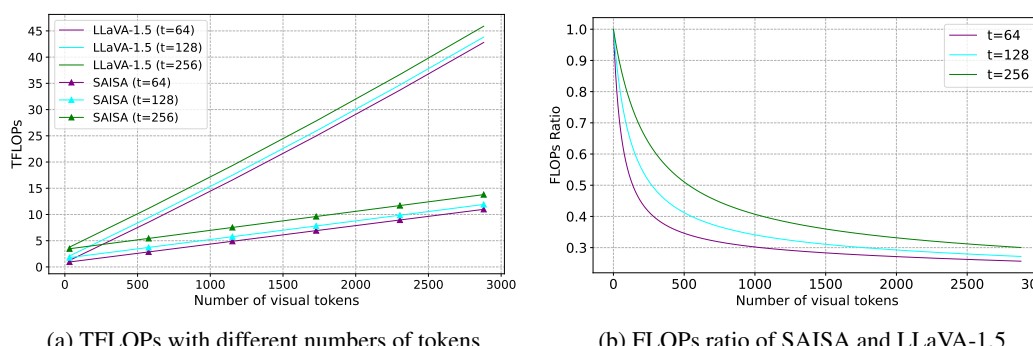

(a) TFLOPs with different numbers of tokens    (b) FLOPs ratio of SAISA and LLaVA-1.5

Figure 7: Inference computational costs comparison with different numbers of visual and text tokens, where t denotes the number of text tokens. SAISA achieves higher computational efficiency.

### 4.3 TRAINING PROCEDURE

The training procedure of SAISA consists of two stages: pre-training and fine-tuning.

**Pre-training.** The objective of pre-training is to transform an LLM into an MLLM with a foundational comprehension of visual information, providing an initialization for the multimodal fine-tuning stage. Following LLaVA-1.5, only the multimodal projector is trainable during this stage. To improve training efficiency, we further reduce the number of trainable parameters in this stage. Specifically, we train a shared MLP for all layers of the LLM.

**Fine-tuning.** The objective of fine-tuning is to enable the model to follow visual instructions from users. As an initialization, we replicate the pre-trained shared MLP $N$ times to initialize the MLPs in the projector, where $N$ denotes the number of layers in the LLM. Following LLaVA-1.5, we utilize visual instruction data to train the model, and both the LLM and the projector are trainable during this stage.

### 4.4 COMPARISON OF COMPUTATIONAL COST

We focus on the computations of the LLM and the projector, because those of the visual encoder are identical in this comparison. For LLM with $n$ layers, we assume $h$ is the hidden state size, $m$ is the intermediate size of the FFNs, $t$ is the number of text tokens, and $v$ is the number of visual tokens. To comprehensively consider LLMs with and without grouped query attention (GQA) (Ainslie et al., 2023), we assume that $k$ is the output dimension of key/value matrices. For the projector, we assume that $d$ is the dimension of the input visual features. The FLOPs of LLaVA-1.5 are $2n(t+v)h(2h+3m+2k)+4n(t+v)^2h+2vhd+2vh^2$ For SAISA, visual tokens are multiplied only by the key and value matrices, and the query sequence length is $t$ in the attention operation. Therefore, the FLOPs are $2nth(2h+3m+2k)+4nvhk+4nt(t+v)h+2nvhd+2nvh^2$. Details of the FLOPs calculation are in the Supplementary Material. Figure 7 compares the FLOPs of SAISA and LLaVA-1.5 with different numbers of tokens, based on Vicuna-7B-v1.5 (Chiang et al., 2023). We can observe that SAISA achieves a higher computational efficiency than LLAVA-1.5 when processing the same numbers of tokens.

## 5 EXPERIMENTS

### 5.1 SETUPS

**Model Configuration.** We mainly compare SAISA with LLaVA-1.5 (Liu et al., 2024a) due to its fully open training data, replicability, and acceptable training costs. We use the same settings as LLaVA-1.5. To be specific, we employ Vicuna-7B-v1.5 (Chiang et al., 2023) as the default LLM backbone and CLIP-ViT-L/14-336 (Radford et al., 2021) as the default visual encoder.

**Training Details.** We utilize the same training data as LLaVA-1.5. The pre-training dataset contains 558k samples, and the fine-tuning dataset contains 665k samples.

| Method | LLM | #Vis. Tok. | Training Data↓ | Inference TFLOPs↓ | MMMU VAL | MME | MMBench EN | MMBench CN | SEED-Bench |
|--------|-----|-----------|---------------|-------------------|----------|-----|------------|------------|------------|
| InstructBLIP | Vicuna-7B | 32 | 130M | 1.25 | 30.6 | 1137.1 | 36.0 | 23.7 | 58.8 |
| MiniGPT-v2 | Llama2-7B | 256 | 326M | 4.20 | 25.0 | 708.4 | 24.3 | - | 29.4 |
| Otter | Llama-7B | 64 | 2.1B | 1.15 | - | 1292.3 | 48.3 | - | 35.2 |
| Shikra | Vicuna-7B | 256 | 6.1M | 4.20 | - | - | 58.8 | - | 44.5 |
| IDEFICS-9B | Llama-7B | 64 | 354M | 1.15 | 18.4 | 942.0 | 48.2 | 25.2 | 44.5 |
| IDEFICS-80B | Llama-65B | 64 | 354M | 11.42 | - | 1244.9 | 54.5 | 38.1 | 53.2 |
| Qwen-VL-Chat | Qwen-7B | 256 | 1.4B | 4.20 | 36.0 | 1435.2 | 60.6 | 56.7 | 65.4 |
| FastV | Vicuna-7B | 576 | 1.2M | 4.90 | 35.8 | - | 64.3 | 56.8 | - |
| VTW | Vicuna-7B | 576 | 1.2M | 4.70 | 36.3 | - | 64.0 | 58.5 | - |
| MQT-LLaVA | Vicuna-7B | 256 | 1.2M | 4.20 | 34.8 | 1434.5 | 64.3 | - | - |
| LLaVA-1.5 | Vicuna-7B | 576 | 1.2M | 8.53 | 35.7 | **1510.7** | 64.3 | 58.3 | **66.1** |
| SAISA (Ours) | Vicuna-7B | 576 | 1.2M | 2.86 | **36.9** | 1461.9 | **65.7** | **59.0** | 64.5 |

Table 2: Performance on comprehensive benchmarks for MLLMs. #Vis. Tok.: the number of visual tokens involved in each image. Training Data: accumulated multimodal pre-training and fine-tuning data volume. ↓: a lower value in these columns is better. Inference TFLOPs: the computational cost of processing a single image when the number of text tokens is 64. Among the baseline models, FastV, VTW and MQT-LLaVA are efficient MLLMs. The comparison between LLaVA-1.5 and SAISA is fair, because they use the same settings and training data.

| Method | LLM | Inference TFLOPs↓ | POPE overall | POPE rand | POPE pop | POPE adv | GQA | ScienceQA IMG | TextVQA | OK-VQA |
|--------|-----|-------------------|--------------|-----------|----------|----------|-----|---------------|---------|--------|
| Shikra | Vicuna-7B | 4.20 | 84.7 | 86.9 | 84.0 | 83.1 | - | - | - | - |
| IDEFICS | Llama-7B | 1.67 | 81.8 | 88.3 | 81.1 | 76.0 | 35.5 | 51.6 | 25.9 | 38.4 |
| IDEFICS | Llama-65B | 16.62 | 77.5 | 86.7 | 74.9 | 70.8 | 45.2 | 61.8 | 30.9 | 45.2 |
| Qwen-VL-Chat | Qwen-7B | 4.20 | 87.0 | **89.0** | 87.4 | 84.7 | 57.5 | 68.2 | **61.5** | 56.6 |
| LLaVA-1.5 | Vicuna-7B | 8.53 | 86.8 | 88.2 | 87.2 | **85.1** | **62.0** | 66.8 | 58.2 | 53.4 |
| SAISA (Ours) | Vicuna-7B | 2.86 | **87.2** | **89.0** | **87.6** | 85.0 | 60.9 | **70.1** | 56.8 | **56.8** |

Table 3: Performance on hallucination and VQA benchmarks. The comparison between LLaVA-1.5 and SAISA is fair.

## 5.2 MAIN RESULTS

The performance on benchmarks is shown in Table 2 and 3. The inference latency test is shown in Table 4. We also include efficient MLLMs in both training-based and training-free manners, such as MQT-LLaVA (Hu et al., 2024), FastV (Chen et al., 2024b) and VTW (Lin et al., 2024).

| Method | 64 Text Tok. Latency (ms) | 128 Text Tok. Latency (ms) | 256 Text Tok. Latency (ms) |
|--------|---------------------------|----------------------------|----------------------------|
| LLaVA-1.5 | 56.0 | 64.4 | 75.6 |
| SAISA (Ours) | 33.0 | 37.1 | 45.9 |

Table 4: Results of the inference latency test.

We evaluate SAISA on a range of benchmarks, including: (1) comprehensive benchmarks for instruction-following MLLMs such as MMMU (Yue et al., 2024), MME (Fu et al., 2023), MM-Bench (Liu et al., 2024b), MMBench-CN (Liu et al., 2024b), and SEED-Bench (Li et al., 2023b); (2) hallucination benchmark such as POPE (Li et al., 2023d), which evaluates MLLMs' degree of hallucination on three subsets: random, popular, and adversarial; (3) general visual question answering benchmarks such as GQA (Hudson & Manning, 2019) and ScienceQA IMG (Lu et al., 2022); (4) fine-grained visual question answering benchmarks such as OK-VQA (Marino et al., 2019) and TextVQA (Singh et al., 2019), OK-VQA requires fine-grained image understanding and spatial understanding, and TextVQA is an OCR-related benchmark; (5) vision-centric MLLM benchmarks such as MMVP (Tong et al., 2024) and CV-Bench (Tong et al., 2025). We report the TFLOPs of processing a single image when the number of text tokens is 64. In the inference latency test, the latency is reported as the time of LLM prefilling during inference with varying numbers of text tokens.

| Method | LLM | Visual Encoder | #Vis. Tok. | Inference TFLOPs↓ | MMMU VAL | MMBench EN | MMBench CN | POPE | GQA | SQA IMG | OK-VQA | Average |
|--------|-----|----------------|-----------|-------------------|----------|-----------|-----------|------|-----|---------|--------|---------|
| LLaVA-1.5 | Vicuna | SigLIP | 729 | 10.63 | 36.6 | 66.2 | 58.9 | 86.5 | 62.5 | **70.5** | **56.4** | 62.5 |
| SAISA (Ours) | Vicuna | SigLIP | 729 | 3.40 | **37.4** | **67.5** | **60.7** | **87.0** | **62.9** | 70.0 | 55.8 | **63.0** |
| LLaVA-1.5 | Vicuna | Conv | 1024 | 14.76 | 34.6 | 56.6 | 49.6 | **88.2** | **61.1** | 66.4 | 51.4 | 58.3 |
| SAISA (Ours) | Vicuna | Conv | 1024 | 4.44 | **35.1** | **61.1** | **54.9** | 87.0 | 57.7 | **66.5** | **54.4** | **59.5** |
| LLaVA-1.5 | Vicuna | CLIP | 576 | 8.53 | 35.7 | 64.3 | 58.3 | 86.8 | **62.0** | 66.8 | 53.4 | 61.0 |
| SAISA (Ours) | Vicuna | CLIP | 576 | 2.86 | **36.9** | **65.7** | **59.0** | **87.2** | 60.9 | **70.1** | **56.8** | **62.4** |
| LLaVA-1.5 | Mistral | CLIP | 576 | 9.17 | 34.8 | 65.9 | 54.9 | **87.2** | **62.0** | **71.6** | 2.5* | 54.1 |
| SAISA (Ours) | Mistral | CLIP | 576 | 2.10 | **35.9** | **67.5** | **57.5** | 86.9 | 61.2 | 71.2 | **23.9*** | **57.7** |
| LLaVA-1.5 | Llama3 | CLIP | 576 | 9.17 | 36.8 | 70.4 | 64.2 | **87.2** | **63.5** | 73.3 | **61.2** | 65.2 |
| SAISA (Ours) | Llama3 | CLIP | 576 | 2.10 | **38.3** | **71.3** | **65.2** | 86.8 | 61.8 | **74.4** | 60.7 | **65.6** |

Table 5: Ablation on LLMs and Visual Encoders. Here, "SigLIP" = SigLIP-ViT-SO400M/14-384, "Conv" = OpenCLIP-ConvNeXt-XXL-1024, and "CLIP" = CLIP-ViT-L/14-336. *Both LLaVA-1.5 and SAISA models on Mistral exhibit low performance on OK-VQA, because they respond to most questions in this benchmark with "Unanswerable".

| Pre-trained Parameters | MMMU VAL | MMBench EN | MMBench CN | POPE | SQA IMG | OK-VQA |
|------------------------|----------|-----------|-----------|------|---------|--------|
| Full Projector | 34.8 | 59.2 | 51.1 | 85.6 | 67.8 | 53.1 |
| Shared MLP | 36.9 | 65.7 | 59.0 | 87.2 | 70.1 | 56.8 |

Table 6: Ablation on Pre-training Strategies.

| Projector | MMMU VAL | MMBench EN | MMBench CN | POPE | SQA IMG | OK-VQA |
|-----------|----------|-----------|-----------|------|---------|--------|
| Linear Layers | 35.7 | 65.3 | 56.6 | 85.8 | 69.2 | 53.6 |
| Resamplers | 35.6 | 59.5 | 50.5 | 83.5 | 69.1 | 53.4 |
| MLPs | 36.9 | 65.7 | 59.0 | 87.2 | 70.1 | 56.8 |

Table 7: Ablation on Projector Designs.

Table 2 shows the comparison on the comprehensive benchmarks. rictSAISA outperforms Instruct-BLIP (Dai et al., 2023), MiniGPT-v2Chen et al. (2023a), Otter (Li et al., 2023a), Shikra (Chen et al., 2023b), and IDEFICS (IDEFICS, 2023) utilizing LLama-7B and LLama-65B (Touvron et al., 2023a) across all these benchmarks. Compared with Qwen-VL-Chat (Bai et al., 2023b) and LLaVA-1.5 (Liu et al., 2024a), SAISA performs better on most benchmarks. Among these baseline models, Otter and IDEFICS employ cross-attention space alignment, whereas the other models utilize embedding space alignment. Table 3 shows the comparison on the hallucination and VQA benchmarks. We also include the comparison on vision-centric benchmarks in the appendix. SAISA achieves the best overall performance compared to the baseline, and strikes a favorable balance between performance and efficiency.

Notably, SAISA inherently supports flexible multi-turn conversations. We include multi-turn results in the appendix.

## 5.3 ABLATION STUDY

**Ablation on LLMs and Visual Encoders.** As presented in Table 5, we perform multiple ablation experiments on both LLM backbones and visual encoders to validate the effectiveness of SAISA. We tune a set of SAISA models using a variety of LLM backbones and visual encoders. The ablated LLMs include Vicuna-7B (Chiang et al., 2023) and two LLMs using grouped query attention (GQA) (Ainslie et al., 2023), such as Mistral-7B (Jiang et al., 2023) and Llama3-8B (Meta, 2024). The ablated visual encoders include two ViT-based (Dosovitskiy et al., 2020) visual backbones such as CLIP-ViT-L/14-336 (Radford et al., 2021) and SigLIP-ViT-SO400M/14-384 (Zhai et al., 2023), and a ConvNeXt-based (Liu et al., 2022) visual encoder such as ConvNeXt-XXL-1024 from Open-CLIP (Ilharco et al., 2021; Schuhmann et al., 2022). The experimental results demonstrate that SAISA consistently achieves superior performance to LLaVA-1.5 across different LLM backbones and visual encoders, while dramatically reducing computational costs.

**Ablation on Pre-training Strategies.** As shown in Table 6, we conduct an ablation to investigate the effects of SAISA's pre-training strategies. We tune a SAISA model where the full projector (32 MLPs) is tunable during pre-training, and the other settings keep the same as the original SAISA. With more randomly initialized parameters, we observe a performance drop when pre-training the full projector. We attribute this drop to the small amount of pre-training data with only 558k samples.

This ablation study demonstrates the effectiveness of our pre-training strategy, which provides a robust initialization for the subsequent fine-tuning stage.

**Ablation on Projector Designs.** Previous studies find that replacing linear projection with MLP projection improves performance in MLLM (Liu et al., 2024a) and self-supervised learning (Chen et al., 2020a;b). We conduct an experiment to investigate the impact of projector designs in SAISA. We tune a model under the same configuration as the original SAISA but replace each MLP in the projector with a linear layer or a resampler (Jaegle et al., 2021). We set the output token number of the resampler projector to 256. Table 7 shows that the model with the MLP projector performs better than the model with other projectors, including the linear projector, which is consistent with the finding of the previous study (Liu et al., 2024a). Furthermore, the linear projector performs better than the resampler projector, because the latter down-samples visual tokens and overlooks fine-grained visual information such as spatial information.

## 6 RELATED WORK

### 6.1 MULTIMODAL LARGE LANGUAGE MODELS

Multimodal Large Language Models (MLLMs) are typically built on Large Language Models (LLMs) (Chiang et al., 2023; Meta, 2024; Jiang et al., 2023; Touvron et al., 2023a;b; Bai et al., 2023a) by aligning visual features generated by visual encoders (Radford et al., 2021; Zhai et al., 2023; Ilharco et al., 2021; Liu et al., 2022) with the LLMs. There are two most common architectures for this purpose, embedding space alignment and cross-attention space alignment. For embedding space alignment (Liu et al., 2023; 2024a; Li et al., 2023c; Dai et al., 2023; Bai et al., 2023b; Chen et al., 2023a; Zhu et al., 2023; Bai et al., 2025; Zhu et al., 2025), MLLMs align visual features with the text token embedding space via a projector and concatenate the visual and text tokens as the LLM input. These models exhibit efficiency during training, but suffer from inference inefficiency because of the long token sequence. For cross-attention space alignment (Alayrac et al., 2022; LAION, 2023; Awadalla et al., 2023), MLLMs introduce new cross-attention blocks for the interaction between text and visual modalities, and align the visual features with the cross-attention spaces. These models achieve efficiency during inference, but require a substantial amount of data to train the new parameters. In this paper, we propose SAISA, an architecture for building MLLMs with efficiency during both training and inference.

### 6.2 VISUAL REDUNDANCY AND EFFICIENCY OPTIMIZATION

Some previous work has discussed redundancy in ViT (Bolya et al., 2022) and MLLMs (Li et al., 2025) and proposed corresponding methods to reduce the computational costs of MLLMs. Notably, Shukor & Cord (2024) expose frozen LLMs to multiple modalities, finds that visual tokens change slightly throughout LLMs, and proposes to skip computations in some FFNs. Chen et al. (2024b) and Lin et al. (2024) reveal the redundancy of visual tokens and prune visual tokens in deep layers, while Shang et al. (2025) and Yang et al. (2025b) prune and merge visual tokens produced by the vision encoder. Yang et al. (2025a) probe the redundancy of visual tokens during inference and explain visual redundancy through language contexts. Unlike previous work, we find that attention among visual tokens is unnecessary in MLLMs, and propose SAISA, a novel architecture to build and train efficient MLLMs. Instead of directly reducing visual token number, SAISA achieves efficiency while maintaining fine-grained visual understanding by preserving the original number of visual tokens.

## 7 CONCLUSION

In this paper, we take a step towards developing MLLMs with efficiency during both training and inference. To achieve this, we conduct a study of current MLLM architectures and find the key factors for building efficient MLLMs. By integrating these factors and gradually reducing redundant computations, we propose **SAISA**, an effective and efficient architecture for MLLMs. SAISA demonstrates the ability to dramatically reduce the computational costs of MLLMs without compromising their capabilities.

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

# A APPENDIX

## A.1 LLM USAGE

We used OpenAI's ChatGPT to help polish the language and improve the readability of the manuscript. Specifically, ChatGPT was used for grammar checking and sentence rephrasing. We list our prompt for using OpenAI's ChatGPT to help polish writing as follows.

---

**Prompt for Using OpenAI's ChatGPT to Help Polish Writing**

```
Below is a paragraph from an academic paper. Polish the
    writing to meet the academic style,improve the spelling,
    grammar, clarity, concision and overall readability.
    Furthermore, list all modification and explain the
    reasons to do so in markdown table. \\
Paragraph: {paragraph}
```

---

## A.2 ANALYSIS OF COMPUTATIONAL OVERHEAD

In this section, we provide details of the calculation of the FLOPs of SAISA and LLaVA-1.5. compare the computational costs of SAISA and LLaVA-1.5. We consider the computations of the LLM and the projector, as the computations of the visual encoder are identical in comparison. We consider the computations of the self-attention blocks and the FFNs in each layer of the LLM.

We denote the hidden state size of the LLM backbone as $h$, and the output dimension of the $W_K$, $W_V$ matrices is $k$. When the token sequence has $l$ tokens, the FLOPs of a vanilla self-attention block contain those of the $W_Q$, $W_K$, $W_V$ and $W_O$ matrices, as well as the self-attention operation. The FLOPs of each of $W_K$, $W_V$ are $2lhk$, and those of each of $W_Q$ and $W_O$ are $2lh^2$, and the FLOPs of the self-attention operation are $4l^2h$. The overall FLOPs of the vanilla self-attention block are $4lh^2 + 4lhk + 4l^2h$. For an feed-forward network (FFN), we assume that their intermediate size is $m$, and the FLOPs are $6lhm$. When the LLM backbone has $n$ layers, the FLOPs are $2nlh(2h + 3m + 2k) + 4nl^2h$.

We use $v$ to denote the number of visual tokens, and $t$ to denote the number of text tokens. For LLaVA-1.5, the visual and text tokens are concatenated and then fed into the LLM backbone, resulting in $v + t$ tokens. As a result, the FLOPs of the LLM backbone are $2n(v + t)h(2h + 3m + 2k) + 4(v + t)l^2h$. For the 2-layer MLP projector, where the hidden layer size is set to $h$, the FLOPs of the first layer are $2vhd$, and those of the second layer are $2vh^2$. The overall FLOPs of LLaVA-1.5 are $2n(v + t)h(2h + 3m + 2k) + 4(v + t)l^2h$.

For SAISA, a NAAViT self-attention block also contains $W_Q$, $W_K$, $W_V$, $W_O$ and the self-attention operation. In contrast to the vanilla self-attention block, the $W_Q$ and $W_O$ matrices are only applied to text tokens in SAISA, and the attention operation among visual tokens are omitted. As a result, the overall FLOPs of a NAAViT self-attention block are $2th(2h + 2k) + 4vhk + 4t(t + v)h$. The FFNs in SAISA are only applied to text tokens, resulting in FLOPs $6thm$. With respect to the projector, which contains $n$ 2-layer MLPs, the FLOPs are $2nvhd + 2nvh^2$. The overall FLOPs of SAISA are $2nth(2h + 3m + 2k) + 4nvhk + 4nt(t + v)h + 2nvhd + 2nvh^2$.

## A.3 IMPLEMENT DETAILS

The detailed training settings and hyperparameters (HPs) of SAISA are summarized in Table 8, including the hyperparameters utilized during the pre-training and fine-tuning. The entire two-stage training process of SAISA is executed on a single node with 8 NVIDIA A800 80G GPUs, employing flash-attention v2 for all experiments.

| Configurations | Pre-training | Fine-tuning |
|---|---|---|
| Projector Init. | Random | Pre-training Stage |
| Training Modules | Shared MLP | Projector, LLM |
| Deepspeed | zero-2 | zero-3 |
| Learning Rate | 1e-3 | 2e-5 |
| Warm-up Ratio | 0.03 | |
| Batch Size | 256 | 128 |
| Bfloat16 | True | |
| LR Schedule | Cosine Decay | |
| Training Steps | 2.2k | 5.2k |
| Weight Decay | 0.0 | |
| Epoch | 1 | |
| Optimizer | AdamW | |

Table 8: Hyperparameters utilized during training. LR schedule denotes learning rate schedule. Init. denotes initialization.

| Method | Inference TFLOPs↓ | MMVP | CV-Bench 2D | CV-Bench 3D |
|---|---|---|---|---|
| LLaVA-1.5 | 8.53 | 24.7 | **56.6** | 59.5 |
| SAISA (Ours) | 2.86 | **26.0** | 56.2 | **59.8** |

Table 9: Performance on vision-centric benchmarks.

## A.4 VISION-CENTRIC BENCHMARKS

In this section, we provide the comparison on vision-centric benchmarks. Since most previous methods do not evaluate their performance on these benchmarks, we compare SAISA with LLaVA-1.5. The results are shown in Table 9.

## A.5 QUALITATIVE RESULTS

In this section, we utilize the qualitative examples in the original LLaVA-1.5 paper to comprehensively demonstrate the abilities of SAISA. The examples, which cover a diverse range of scenarios and challenges, are shown in Table 12, 11, 13 and 14. For comparison, we prompt queries in these examples to LLaVA-1.5 and include its response in most examples. Our SAISA, trained on the same data as LLaVA-1.5, demonstrates similar performance across various unseen response format prompts, while exhibiting superior training and inference efficiency. It can support flexible multi-turn conversations (Table 10) and accurately identify factual inaccuracies in questions and provide more effective and reasonable responses to complex queries (Table 12). SAISA also generates fine-grained prompts for Stable Diffusion models in a specified format (Table 11). Furthermore, it can offer responses with adjustable levels of detail based on user needs, adapting its output across multiple dialogue rounds (Table 13), and it can perform detailed writing tasks based on the given image (Table 14).

**Multi-turn Conversation:**

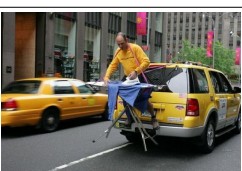

| | |
|---|---|
| User | What is unusual about this image? **Answer briefly with a single sentence.** |
| LLaVA-1.5 | A man is hanging clothes on a clothesline attached to a car. |
| SAISA | A man is ironing clothes while standing on the back of a taxi. |
| User | **Give a more detailed explanation.** |
| LLaVA-1.5 | A man is hanging clothes on a clothesline attached to a car, which is an unusual sight as people typically hang clothes on a clothesline at home or in a designated area. |
| SAISA | The unusual aspect of this image is that a man is ironing clothes while standing on the back of a taxi. This is not a typical scene, as ironing clothes is usually done indoors, and standing on the back of a moving vehicle is not a safe or practical way to do it. The man's actions may be a stunt or a publicity stunt, but it is certainly an unconventional and potentially dangerous situation. |

Table 10: SAISA can dynamically adjust response granularity across multiple dialogue rounds.

**Visual input example, Constrained SD Prompt Generation:**

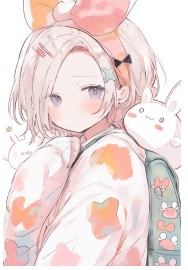

| | |
|---|---|
| User | Here's a formula for a Stable Diffusion image prompt: a cartoon-style image of [adjective] [subject] [haircut] [clothing] [doing action], [color scheme], detailed, trending on artstation. Generate the prompt for this image. |
| LLaVA-1.5 | A cartoon-style image of a young girl with a heart-shaped bow in her hair, wearing a white and pink jacket, and holding a stuffed bunny. The image has a watercolor color scheme, and it is detailed and trending on ArtStation. |
| SAISA | A cartoon-style image of a young girl with a cute haircut, wearing a white jacket and a bow, holding a stuffed animal and a backpack. The image has a color scheme of pink, white, and orange, and is detailed and trending on ArtStation. |

Table 11: SAISA can generating prompts in a specified format tailored for Stable Diffusion models.

**Visual input example, Tricky Question:**

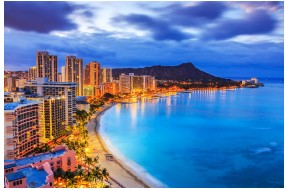

| | |
|---|---|
| User | If there are factual errors in the questions, point it out; if not, proceed to answering the question. What's happening in the desert? |
| LLaVA-1.5 | There is no desert in the image. The image features a beach with a city skyline in the background, and a large body of water. |
| SAISA | In the image, there is no desert visible. The scene takes place on a beach with a city skyline in the background. The beach is located near a city, and there is a lively atmosphere with lights reflecting on the water. |

Table 12: SAISA can evaluate whether the question contains factual inaccuracies and respond to challenging questions.

| | |
|---|---|
| **Visual input example, Different Format Prompts:** | |

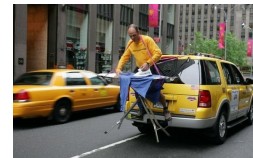

| | |
|---|---|
| User | What is the color of the shirt that the man is wearing? **Answer the question using a single word or phrase.** |
| LLaVA-1.5 | Yellow |
| SAISA | Yellow |
| User | Is this unusual? **Please explain briefly with a short sentence.** |
| LLaVA-1.5 | Yes, it is unusual for a man to be drying clothes on the back of a moving vehicle. |
| SAISA | Yes, it is unusual for a man to be ironing clothes while standing on the back of a taxi. |
| User | Is this unusual? **Please explain in detail.** |
| LLaVA-1.5 | Yes, this is an unusual scene. In the image, a man is sitting on a folding chair in the back of a yellow taxi, holding a clothesline with clothes on it. This is not a typical sight in a city setting, as people usually do not transport laundry in such a manner. It is also uncommon to see someone sitting on a folding chair in the back of a taxi, as passengers usually sit inside the vehicle. The combination of the man's unusual activity and the unconventional seating arrangement makes this scene stand out as an atypical and intriguing moment. |
| SAISA | Yes, this is an unusual scene. In the image, a man is standing on the back of a yellow taxi, holding a clothes iron. This is not a typical sight, as people usually do not use taxis for ironing clothes. It is also not common to see someone ironing clothes in public, as it is generally done indoors in a private setting, such as a home or an office. The man's actions in the image are unconventional and seem to be a humorous or artistic representation of an everyday situation. |

Table 13: SAISA is capable of providing responses at various levels of granularity according to user requirements.

**Visual input example, Visual Writing Task:**

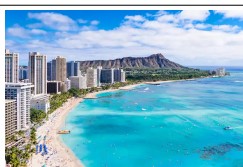

| | |
|---|---|
| User | Compose an engaging travel blog post about a recent trip to this place, highlighting cultural experiences and must-see attractions. |
| SAISA | Title: Paradise Found: A Trip to the Hawaiian Islands |

Introduction:

I recently had the opportunity to visit the beautiful Hawaiian Islands, and I must say, it was an unforgettable experience. The breathtaking views, crystal blue waters, and lush green landscapes truly make it a paradise on earth. In this blog post, I'll share some of my favorite moments and must-see attractions from my trip.

Day 1: Arrival and Exploring Oahu

Upon arrival, I was greeted by the stunning view of Waikiki Beach, which is located on the island of Oahu. I spent the day exploring the area, taking in the sights, and soaking up the sun. One of the highlights of my day was visiting the iconic Diamond Head, a volcanic crater that offers panoramic views of the beach and ocean.

Day 2: Polynesian Cultural Center

The next day, I visited the Polynesian Cultural Center, where I had the chance to experience the rich culture and traditions of various Polynesian islands. The center showcases traditional dances, music, and crafts, as well as interactive exhibits that allowed me to learn more about the history and customs of the region.

Day 3: Hanauma Bay

On my third day, I visited Hanauma Bay, a stunning natural wonder that is home to an abundance of marine life. The bay is a protected marine sanctuary, and visitors are required to take a guided tour to learn about the local ecosystem and conservation efforts. The crystal blue waters and vibrant coral reefs were truly breathtaking, and I was able to witness sea turtles, colorful fish, and other marine creatures up close.

Day 4: Pearl Harbor and USS Arizona

On my last day, I visited Pearl Harbor, the site of the infamous attack that led to the United States' entry into World War II. The USS Arizona Memorial is a must-see attraction, as it offers a solemn and educational experience. I was able to learn about the history of the attack and the significance of the USS Arizona in the war.

Conclusion:

In conclusion, my trip to the Hawaiian Islands was an unforgettable experience filled with cultural experiences, breathtaking views, and adventure. From the iconic Waikiki Beach to the protected marine sanctuary of Hanauma Bay, there is truly something for everyone to enjoy. I would highly recommend a visit to these beautiful islands for anyone looking to experience the wonders of nature and the rich history of the region.

Table 14: SAISA is capable of executing complex writing tasks based on a given image.

