# OpenReview forum: "SAISA: Towards Multimodal Large Language Models with Both Training and Inference Efficiency"
_ICLR.cc/2026/Conference — Submitted to ICLR 2026_

### Official Review · Reviewer_QKS9 · 2025-10-25

**Soundness:** 3
**Presentation:** 3
**Contribution:** 2
**Rating:** 4
**Confidence:** 4

**Summary:**

The paper proposes an approach to improve the training and inference efficiency of MLLMs by exploiting redundancy among visual tokens. The key idea is to eliminate the attention computations and FFNs operations related to visual token. The proposed architecture demonstrates significant computational savings with competitive performance to baselines. Experiments across multiple benchmarks, vision encoders, and LLM backbones validate the approach.

**Strengths:**

* The paper tackles an important problem: improving the efficiency of MLLMs, which are known to be computationally demanding.
* The writing is clear, well structured and the work is well motivated.
* The approach is validated across different architectures and setups

**Weaknesses:**

* The redundancy among visual tokens has been explored in prior work. The paper should better position itself in relation to existing studies that analyze or reduce visual redundancy e.g.  [1].

* The experiments primarily use the LLaVA setup, which lags behind more recent MLLM work. It is unclear how the proposed method compares with state-of-the-art (SOTA) models or whether it can be integrated effectively into stronger baselines.

* The paper does not discuss scaling behavior—how the proposed method performs when model or dataset size increases (also related to the previous point).

* Many modern VLM benchmarks (e.g., OCR, document understanding, PDF reasoning) require high-resolution images and fine-grained visual features. It remains unclear whether removing attention among visual tokens will negatively affect performance in these scenarios.

[1] "Implicit multimodal alignment: On the generalization of frozen llms to multimodal inputs." NeurIPS2024.

**Questions:**

Please refer to the weaknesess section.

---

> ### Author Response · Authors · 2025-11-23
> **Author Response**
>
> Thank you for providing your valuable and constructive feedback! Please find below the responses to each comment.
> ***
>
>
> **[Weakness 1] Comparison with prior work related to the redundancy among visual tokens.**
>
> Thank you for the helpful pointer. We agree that prior work examining implicit alignment and modalities, such as [1], is relevant and should be explicitly discussed. We read the NeurIPS paper and find the paper highly insightful. Some of their empirical observations help to interpret our findings: for example, perceptual tokens change slightly throughout the model and they propose to skip computations in some FFNs. However, our paper focuses on a different, complementary question — whether attention among visual tokens inside the LLM is necessary for MLLMs. We removes all attention among visual tokens and substitutes lightweight MLPs for all visual token processing instead of FFNs across the LLM. **Our contribution is a new architecture to build and train efficient MLLMs** other than the typical embedding space alignment architecture. Our SAISA introduces a different way of eliminating redundant computation and achieves substantial **efficiency gains together with improved performance**. We have included [1] and other work about redundancy among visual tokens, such as [2] etc, as related work in Subsection 6.2 of the revised paper and comparison to them.
>
> [1] "Implicit multimodal alignment: On the generalization of frozen llms to multimodal inputs." NeurIPS2024.
> [2] Skipping computations in multimodal llms.
>
> ***
>
> **[Weakness 2 & Weakness 3] Experiments on stronger models, scaled dataset size, and scaled model size.**
>
> The purpose of our paper is to introduce a new architecture, improving computational efficiency compared to current mainstream embedding space alignment architecture. To make a comparison between two architectures, it's important to use identical experimantal settings, e.g. same training data and training settings. However, **it is impractical for us to make a fair comparison between our method and current SOTA MLLMs (e.g. Qwen series), because we don't have enough computing resources, and the training data of these models is not open source**. We note that current SOTA MLLMs, including Qwen series, still use the embedding space alignment architecture. Since LLaVA-1.5 is a crucial baseline of this architecture, the comparison with LLaVA-1.5 is representative. To further prove the effectiveness of our method, we integrate it into **EAGLE (ICLR25)** [3], a **stronger baseline** with **more advanced architecture designs** and **scaled dataset size. More importantly, it's training dataset is fully open source**. We present the experimental results as follows.
>
> | Method | MMMU | MMBench-EN | MMBench-CN | POPE | ScienceQA | GQA |
> |--------|------|-------------|-------------|------|-----------|------|
> | EAGLE | 36.3 | 68.4 | 61.5 | 88.5 | 70.0 | 64.9 |
> | Ours | 37.1 | 68.8 | 62.1 | 88.8 | 71.0 | 64.8 |
>
> We observe that our method achives better overall performance, which demonstrates that it can be integrated effectively into stronger models.
>
> Other than conducting experiments on the scaled EAGLE dataset, we also conduct experiments on scaled model size. **We scale the model size to 13B**. We present the experimental results as follows. We observe that our method also performs well when model size increases.
>
> | Method | MMMU | MMBench-EN | MMBench-CN | POPE | ScienceQA | GQA |
> |-----------------|------|------------|------------|------|-----------|------|
> | LLaVA-1.5-13B | 35.3 | 67.7 | 63.6 | 87.1 | 71.6 | 63.3 |
> | SAISA-13B | 35.9 | 68.1 | 64.1 | 87.3 | 71.7 | 62.9 |
>
> [3] Eagle: Exploring The Design Space for Multimodal LLMs with Mixture of Encoders
>
> ***
>
> **[Weakness 4] Experiments on benchmarks that require high-resolution images and fine-grained visual features.**
>
> We conduct evaluation on VLM benchmarks that require high-resolution images and fine-grained visual features, such as TextVQA, AI2D, OCRBench, ChartQA and DocVQA. The following table presents a comparison between vanilla LLaVA-1.5 and our method that removes attention among visual tokens. From the experimental results, we observe that our method still achieves similar performance on these benchmarks.
>
> | Method | TextVQA | AI2D | OCRBench | ChartQA | DocVQA |
> |------------|---------|------|----------|---------|--------|
> | LLaVA-1.5 | 58.2 | 54.8 | 31.3 | 18.2 | 21.5 |
> | Ours | 58.1 | 55.2 | 31.3 | 18.2 | 21.4 |

---

### Official Review · Reviewer_Cqp8 · 2025-10-27

**Soundness:** 3
**Presentation:** 3
**Contribution:** 3
**Rating:** 4
**Confidence:** 4

**Summary:**

This paper addresses the well-known trade-off between training and inference efficiency in MLLMs. The authors categorize existing architectures into two types: embedding space alignment (e.g., LLaVA), which is training-efficient but inference-inefficient due to long visual token sequences, and cross-attention space alignment (e.g., Flamingo), which is inference-efficient but training-inefficient due to a large number of new parameters.

To tackle this, the paper first proposes NAAViT, a self-attention mechanism that eliminates the computationally expensive attention among visual tokens. A pilot study suggests this form of attention is redundant. Building on this, the authors introduce SAISA (Self-Attention Input Space Alignment), a novel architecture that employs NAAViT blocks and aligns visual features directly with the input spaces of each self-attention layer in the LLM. This approach avoids both the quadratic complexity of visual self-attention and the need to pass visual tokens through the FFNs. The experimental results demonstrate that SAISA significantly reduces inference FLOPs (by 66%) and training costs (by 26%) compared to its primary baseline, LLaVA-1.5, while achieving competitive or superior performance on several benchmarks.

**Strengths:**

1. Well-Motivated Problem: The paper targets a critical and practical challenge in the MLLM domain. Improving both training and inference efficiency is a highly valuable research direction.
2. Intuitive and Simple Core Idea: The central concept of NAAViT—that attention between visual tokens within the LLM might be redundant—is simple to understand and implement. The pilot experiment provides good initial evidence for this hypothesis.
3. Strong Empirical Efficiency Gains: The reported improvements in computational efficiency are substantial. A 66% reduction in inference FLOPs and a 26% reduction in training GPU hours compared to LLaVA-1.5 are impressive results that clearly demonstrate the practical benefit of the proposed architecture.
4. Clear Presentation: The paper is well-written, and the diagrams (especially Figure 2 and 3) effectively illustrate the differences between existing architectures and the proposed SAISA model.

**Weaknesses:**

1. While the combination of components is new, the core ideas feel more incremental than foundational.
The primary mechanism, NAAViT, essentially modifies the self-attention in embedding-space models to mimic the behavior of cross-attention models like Flamingo, where text queries attend to visual keys/values without any V-V interaction. The novelty lies in achieving this within a standard self-attention block via masking, rather than by introducing new cross-attention modules, thereby saving parameters. However, the conceptual leap is not substantial.
The idea of projecting features into multiple layers of a transformer is also not entirely new and has precedents in other architectures that seek to inject conditioning information throughout a network. The contribution seems to be a clever engineering combination of existing principles rather than a fundamentally new architectural paradigm.
2. The paper's ablations are too limited to fully substantiate its claims and disentangle the sources of improvement.
The most critical missing ablation is a direct comparison between a model using only NAAViT (as in the pilot experiment) and the full SAISA model (NAAViT + layer-wise projection). The current presentation conflates the benefits of two distinct architectural changes. It is unclear how much of the performance gain and efficiency improvement comes from simply removing visual self-attention versus the more complex layer-wise projection scheme. A thorough analysis is needed to justify the added complexity of SAISA's projector.
The paper argues for projecting visual features to every layer. Is this necessary? An ablation studying the impact of projecting to only a subset of layers (e.g., early, middle, or late layers) would provide deeper insights into how and where visual information is most effectively integrated.
3. The experimental comparison, while thorough against LLaVA-1.5, feels dated and lacks engagement with the latest state-of-the-art in efficient MLLMs.
The MLLM field is advancing at an extremely rapid pace. While LLaVA-1.5 is a crucial baseline, many newer and more efficient models have been proposed since its release. The comparison set in Table 2 feels somewhat stale, missing more recent architectures that have been published in late 2024 or 2025 (as this paper targets ICLR 2026).
More importantly, the paper does not adequately compare its architectural approach to orthogonal efficiency methods, particularly token pruning/merging. Models like FastV (which is cited but could be compared more directly) or other token reduction techniques achieve inference efficiency by dynamically removing visual tokens. SAISA's approach is to keep all tokens but reduce computation per token. A direct and fair comparison between these two competing strategies for efficiency is essential to contextualize SAISA's contribution. Is it better to prune tokens or to keep them all with cheaper attention? This question is left unanswered.

**Questions:**

See above

---

> ### Author Response · Authors · 2025-11-24
> **Author Response (1/n)**
>
> Thank you for providing your valuable and constructive feedback! Please find below the responses to each comment.
>
> ***
>
> **[Weakness 1] Novelty clarification.**
>
> Thank you for your concern about our work. SAISA is built upon findings of prior work, but we would like to clarify why the contribution goes beyond a mere engineering combination and provides new architectural insights.
>
> 1. Although cross-attention models (e.g. Flamingo) also use text tokens as queries to attend to visual keys/values, **existing methods achieve this only by adding cross-attention modules**, which are **trained from scratch** and therefore require substantial additional parameters and large amounts of multimodal data. Previous work [1] demonstrates that **cross-attention models perform worse than embedding-space models** when trained on the same data. In contrast, our work demonstrates that the **pre-trained self-attention blocks of an LLM can natively support this interaction pattern**. Importantly, because the LLM’s attention blocks have already been extensively pre-trained on large-scale data, **their parameters are already highly capable of modeling complex attention relationships**. Leveraging these mature attention modules—rather than introducing new, randomly initialized cross-attention parameters—greatly reduces the training burden. Our preliminary experiment demonstrates that **NAAViT outperforms embedding-space models when using identical training settings**. Moreover, we add a new subsection, Subsection 3.2 Redundancy Analyses, in the revised paper to analyze redundancy in current embedding-space models, providing more evidence for the effectiveness of NAAViT.
>
> 2. For the idea of projecting features into multiple layers of a transformer, **our method is also different from other architectures**. For example, cross-attention models (e.g. Flamingo) project features into cross-attention blocks. SAISA uses these projections in a different way: each projection aligns to the **self-attention input spaces** of the LLM. Moreover, the choice of using **MLPs** rather than linear layers is also well motivated: after we apply NAAViT (in Section 3), a visual tokens only passes through multiple FFNs (MLPs) in the LLM backbone, and it doesn't interact with other tokens. More specifically, the processing of the visual token X can be formulated as:
> $Y_i = \text{FFN}_i(\dots \text{FFN}_2(\text{FFN}_1(X)) \dots),$
> only involving X, and no other tokens. Then we can simplify the transformation of multiple MLPs to a single 2-layer MLP, which results in SAISA. In brief, **we use MLPs to mimic the transformations in multiple FFNs of embedding-space models**.
>
> 3. Our work yields a new alignment paradigm that **simultaneously preserves training efficiency and achieves inference efficiency**. **Prior works either improved one side of the trade-off but not both.**
>
> In summary, SAISA is not only an engineering combination of existing components. It introduces a new understanding of redundancy in multimodal self-attention, a novel self-attention mechanism (NAAViT) that re-purposes pre-trained sel-attention blocks without adding new modules, and an alignment strategy that **unifies the benefits of embedding-space and cross-attention models**. We believe these findings constitute a meaningful conceptual and architectural contribution.
>
> [1] Nvlm: Open frontier-class multimodal llms.

---

> ### Author Response · Authors · 2025-11-24
> **Author Response (2/n)**
>
> **[Weekness 2] Limited ablations.**
>
> We present a direct comparison between the model using only NAAViT and the full SAISA model as follows.
>
> | Model | TFLOPs | MMMU | MMBench-EN | MMBench-CN | POPE | ScienceQA | OKVQA |
> |-----------|--------|------|------------|------------|------|-----------|-------|
> | LLaVA-1.5 | 8.53 | 35.7 | 64.3 | 58.3 | 86.8 | 66.8 | 53.4 |
> | NAAViT | 7.10 | 36.0 | 64.9 | 58.4 | 86.9 | 68.7 | 56.0 |
> | SAISA | 2.86 | 36.9 | 65.7 | 59.0 | 87.2 | 70.1 | 60.9 |
>
> NAAViT reduces FLOPs by 17%, while SAISA achieves a 66% reduction in FLOPs. This implies that the layer-wise projection accounts for 49% reduction in FLOPs, representing a substantial portion of the overall efficiency improvement, although it adds some complexity.
>
> However, we emphasize that although layer-wise projection contributes more to efficiency, the effectiveness of NAAViT is the foundation that enables us to use the layer-wise projection on visual tokens. As discussed in our previous response Author Response (1/n), layer-wise projection is used to mimic the transformations on visual tokens performed by multiple FFNs in embedding-space models. The reason we can do this simplification is the effectiveness of NAAViT. After applying NAAViT, the model’s transformations on each visual token become independent of other tokens.
>
> Regarding the performance comparison, we find that **using NAAViT alone outperforms vanilla LLaVA-1.5**. We attribute such improvement to the following reason: LLaVA-1.5 apply causal masking to both text and visual tokens. While texts are inherently sequential, images are two-dimensional. Consequently, imposing such attention to visual tokens introduces unnecessary noise. Moreover, **SAISA achieves better performance than using NAAViT alone**. This improvement arises because: when using NAAViT alone, we apply FFNs to both visual and text tokens. Since these FFN parameters in the LLM backbone are pre-trained on a large scale of text data, they are optimized for processing text tokens. Although we then train them on both visual and text tokens, as the two modalities demonstrate a clear modality gap in the embedding space [1], and the scale of optimized visual tokens is typically smaller than text tokens, these parameters cannot effectively handle both modalities. We believe this imbalance is also a reason of the "language-prior" phenomenon. In contrast, SAISA uses other MLPs to process visual tokens, and FFNs are optimized to process only text tokens.
>
> Moreover, following your insightful suggestion, we conduct an ablation to study the impact of projecting to only a subset of layers, such as early layers (layer 1,2,3,4,5,6,7,8,9,10,11,12,13,14,15,16), middle layers (layer 9,10,11,12,13,14,15,16,17,18,19,20,21,22,23,24), and late layers (layer 17,18,19,20,21,22,23,24,25,26,27,28,29,30,31,32).
>
> | Layers | MMMU | MMBench-EN | MMBench-CN | POPE | ScienceQA | GQA |
> |--------|------|------------|------------|------|-----------|-----|
> | Early | 36.8 | 65.6 | 58.9 | 87.1 | 70.0 | 59.3 |
> | Middle | 35.5 | 65.2 | 57.2 | 86.1 | 69.4 | 58.4 |
> | Late | 35.2 | 64.3 | 57.0 | 85.0 | 69.2 | 56.8 |
> | All | 36.9 | 65.7 | 59.0 | 87.2 | 70.1 | 60.9 |
>
> The results show that **projecting to all layers gives the best performance**, and **adding visual features to early layers works better than middle or late layers**, demonstrating **visual information is most effectively integrated in early layers**. This observation **aligns with the findings in embedding-space models**. For example, previous work, e.g. FastV, finds that **visual tokens in early layers are more important than those in deeper layers**.
>
> [1] Deciphering cross-modal alignment in large vision-language models with modality integration rate.

---

> ### Author Response · Authors · 2025-11-24
> **Author Response (3/3)**
>
> **[Weakness 3] Experiments on newer models and comparison with orthogonal efficiency methods.**
>
> To make a comparison between model architectures, it's important to use identical experimental settings, e.g. same training data and training settings. It is impractical for us to make a fair comparison between our method and many SOTA MLLMs (e.g. Qwen series), because we don't have enough computing resources, and the training data of these models is not open source. To further prove the effectiveness of our method, we integrate it into EAGLE [1], published at ICLR 2025 (April 2025). EAGLE is a stronger baseline with more advanced architecture designs. More importantly, it's training dataset is fully open source. We present the experimental results as follows. Our method achieves better overall performance, which demonstrates that it can be integrated effectively into stronger baselines.
>
> | Method | MMMU | MMBench-EN | MMBench-CN | POPE | ScienceQA | GQA |
> |--------|------|-------------|-------------|------|-----------|------|
> | EAGLE | 36.3 | 68.4 | 61.5 | 88.5 | 70.0 | 64.9 |
> | Ours | 37.1 | 68.8 | 62.1 | 88.8 | 71.0 | 64.8 |
>
> Regarding the comparison with orthogonal efficiency methods, **we kindly note that we have already provided a direct comparison with FastV and VTW (token pruning mthods) in Table 2 of our original paper**. To make more comprehensive comparison, we include **many newer orthogonal efficiency methods**, such as TopV (CVPR 2025) [2], Libra-Merging (CVPR 2025) [3], VisionZip (CVPR 2025) [4], SparseVLM (ICML 2025) [5], PAR (ICCV 2025) [6], VisiPruner (EMNLP 2025) [7], expanding both token pruning and token merging. The results are as follows.
>
> | Method | TFLOPs | MMMU | MMBench-EN | MMBench-CN | POPE | ScienceQA | GQA |
> |---------------|--------|------|------------|------------|-------|-----------|------|
> | FastV | 4.90 | 35.8 | 64.3 | 56.8 | 82.45 | 67.8 | 60.2 |
> | VTW | 4.70 | 36.3 | 64.0 | 58.5 | 86.0 | 69.1 | 55.1 |
> | TopV | 4.35 | 35.1 | 59.7 | - | 84.2 | 69.6 | - |
> | Libra-Merging | 3.16 | - | 63.9 | 58.2 | - | 69.2 | 60.7 |
> | VisionZip | 2.84 | 36.2 | 64.3 | 56.7 | 84.9 | 68.2 | 60.1 |
> | SparseVLM | 2.84 | 33.8 | 60.0 | 51.1 | 80.5 | 67.1 | 59.5 |
> | PAR | 4.00 | - | 60.2 | 50.7 | 73.6 | 69.6 | - |
> | VisiPruner | 3.93 | - | 62.0 | - | 84.4 | 66.7 | 60.3 |
> | SAISA | 2.86 | 36.9 | 65.7 | 59.0 | 87.2 | 70.1 | 60.9 |
>
> We observe that **SAISA consistently outperforms token reduction techniques**. Moreover, we note that **our method and most token reduction methods can be combined**. For example, we apply VTW to SAISA, and the experimental results are as follows. Combining both methods further improves MLLM efficiency, and SAISA + VTW outperforms LLaVA-1.5 + VTW, demonstrating **strong compatibility between SAISA and token reduction**.
>
> | Method | TFLOPs | MMMU | MMBench-EN | MMBench-CN | POPE | ScienceQA | GQA |
> |------------------|--------|------|------------|------------|------|-----------|------|
> | LLaVA-1.5 + VTW | 4.70 | 36.3 | 64.0 | 58.5 | 86.0 | 69.1 | 55.1 |
> | SAISA | 2.86 | 36.9 | 65.7 | 59.0 | 87.2 | 70.1 | 60.9 |
> | SAISA + VTW | 1.85 | 36.7 | 65.6 | 58.8 | 87.0 | 70.2 | 57.7 |
>
> [1] Eagle: Exploring The Design Space for Multimodal LLMs with Mixture of Encoders.
>
> [2] TopV: Compatible Token Pruning with Inference Time Optimization for Fast and Low-Memory Multimodal Vision Language Model.
>
> [3] Libra-Merging: Importance-Redundancy and Pruning-Merging Trade-Off for Acceleration Plug-In in Large Vision-Language Model.
>
> [4] VisionZip: Longer is Better but Not Necessary in Vision Language Models.
>
> [5] SparseVLM: Visual Token Sparsification for Efficient Vision-Language Model Inference.
>
> [6] Pruning All-Rounder: Rethinking and Improving Inference Efficiency for Large Vision Language Models.
>
> [7] VisiPruner: Decoding Discontinuous Cross-Modal Dynamics for Efficient Multimodal LLMs.

---

### Official Review · Reviewer_JnP5 · 2025-10-27

**Soundness:** 1
**Presentation:** 2
**Contribution:** 1
**Rating:** 2
**Confidence:** 4

**Summary:**

This paper addresses the trade-off between training and inference efficiency in MLLMs. Traditional embedding space alignment models (e.g., LLaVA) are efficient to train but slow to infer, while cross-attention space alignment*models (e.g., Flamingo) offer fast inference but incur high training costs. After analyzing the computational bottlenecks of both paradigms, the authors argue that self-attention among visual tokens is largely redundant. They propose NAAViT to remove visual–visual attention and build upon it to design SAISA, which maps visual features directly to the input spaces of self-attention layers in the LLM while skipping FFN operations on visual tokens. Experiments show that SAISA achieves a 26% reduction in training cost and a 66% reduction in inference FLOPs, with slightly improved accuracy—thus realizing efficiency in both training and inference.

**Strengths:**

1. **Clear problem statement and method design.** The paper tackles the key challenge of improving MLLM computational efficiency—a crucial factor for deployment and energy optimization. The combination of NAAViT’s `removal of visual–text attention` and SAISA’s `layer-wise input-space alignment` provides a novel trade-off between existing embedding- and cross-attention-based architectures.

2. **Fair experimental comparison.** All main experiments are conducted under identical data and training settings as LLaVA-1.5, ensuring fair comparison. The ablations demonstrate generality across different LLMs (Vicuna, Llama3, Mistral) and visual encoders (CLIP, SigLIP, ConvNeXt), while also studying projector design and pretraining strategies.

3. **Strong reproducibility.**  The paper includes detailed hyperparameters, training settings, and FLOPs derivations, providing sufficient information for replication.

**Weaknesses:**

1. **Insufficient evidence for the `redundancy` claim.** The paper concludes that attention among visual tokens is redundant, but this is supported only by a marginal performance gain of NAAViT over standard self-attention. Such improvement could stem from optimization dynamics or implicit regularization rather than true redundancy. This would be meaningful: Providing attention heatmaps or token correlation analyses, and testing on multi-object or spatial reasoning tasks would help verify whether visual–visual interactions are indeed dispensable.

2. **Unclear source of efficiency gains.** The study does not isolate the contributions of NAAViT versus the removal of FFNs on visual tokens. Without reporting FLOPs and performance when using NAAViT alone, it is unclear which component drives the observed acceleration.

3. **Potential incompatibility of NAAViT masks with inference accelerators.** In multi-image or interleaved dialogue scenarios, NAAViT introduces fragmented attention masks that may reduce the efficiency of Flash-Attention 2 or vLLM kernels.

4. **Outdated baseline selection for token compression.** Easy token unshuffle methods can reduce token counts to 1/4 or 1/9 with negligible information loss. Yet the paper lacks comparison with recent **parameter-free token optimization methods**, and only contrasts with older pruning-based approaches from 2024 (e.g., FastV, VTW), which weakens its claims.

5. **Limited experimental scope.** All primary results are based on Vicuna-7B and LLaVA datasets, without evaluation on larger models (e.g., 13B) or other modalities such as videos and multi-image inputs.

**Questions:**

1. Could the authors provide results using NAAViT alone (while keeping FFNs for visual tokens) to disentangle the sources of efficiency?
2. Does the sparse attention mask in NAAViT affect compatibility or speedup when using Flash-Attention 2 or vLLM inference engines, especially for interleaved or long-context inputs?
3. See `Weaknesses` for additional clarifications.

---

> ### Author Response · Authors · 2025-11-23
> **Author Response (1/n)**
>
> Thank you for providing your valuable and constructive feedback! Please find below the responses to each comment.
> ***
>
> **[Weakness 1] Insufficient evidence for the redundancy claim.**
>
> Thank you for the insightful comment. To address the concern that the evidence for the redundancy claim was insufficient, we have added a new subsection, **Subsection 3.2 Redundancy Analyses**, in the revised paper. This new subsection introduces several complementary analyses that strengthen our evidence:
>
> 1. **Cosine similarity-based token hidden states analysis.** We measure the transformation strength that each self-attention block applies to visual tokens by computing the cosine similarities between the input and output tokens at each block. The results show that self-attention block in shallow layers apply almost no transformation to visual tokens, indicating strong redundancy at the early stages of the model.
>
> 2. **Attention map visualizations.** We visualize attention heatmaps. In deeper layers—those directly influencing the final prediction—visual tokens receive extremely little attention from model's output tokens. This suggests that even if some transformations occur on visual tokens in deeper layers, they have almost no downstream influence.
>
> 3. **KL-divergence–based ablations.** To quantify contribution more directly, we remove each attention block’s transformations on visual tokens (while keeping text-token transformations intact) and compute the KL divergence between the modified model’s output logits and those of the original model. Across all layers, ablating transformations on visual tokens yields consistently lower KL divergence than ablating those on text tokens, showing that visual-token updates have minimal impact on the model’s output.
> Together, these analyses provide more direct and quantitative support for our claim, thereby reinforcing the motivation behind NAAViT.
>
> We also test our method on multi-object and spatial reasoning tasks, including GQA, Refcoco and Refcoco+. The results are presented below, demonstrating that visual–visual interactions are indeed dispensable.
> | Model | GQA | Refcoco | Refcoco+ |
> |-----------|------|---------|----------|
> | LLaVA-1.5 | 62.0 | 29.8 | 28.9 |
> | NAAViT | 62.4 | 30.2 | 29.1 |
>
> ***
> **[Weakness 2 & Question 1] Unclear source of efficiency gains.**
>
> Following your suggestion, we present a direct FLOPs and performance comparison between NAAViT and SAISA as follows.
>
> | Model | TFLOPs | MMMU | MMBench-EN | MMBench-CN | POPE | ScienceQA | OKVQA |
> |-----------|--------|------|------------|------------|------|-----------|-------|
> | LLaVA-1.5 | 8.53 | 35.7 | 64.3 | 58.3 | 86.8 | 66.8 | 53.4 |
> | NAAViT | 7.10 | 36.0 | 64.9 | 58.4 | 86.9 | 68.7 | 56.0 |
> | SAISA | 2.86 | 36.9 | 65.7 | 59.0 | 87.2 | 70.1 | 60.9 |
>
> NAAViT reduces FLOPs by 17%, while SAISA achieves a 66% reduction. This implies that the removal of FFNs accounts for 49% reduction in FLOPs, representing a substantial portion of the overall efficiency gains.
>
> However, we want to clear that although the removal of FFNs contributes the most to the efficiency gains, the effectiveness of NAAViT is the foundation that enables us to remove FFNs on visual tokens. After we apply NAAViT, a visual tokens only passes through multiple FFNs (MLPs) in the LLM backbone, and it doesn't interact with other tokens. More specifically, the processing of the visual token X can be formulated as:
>
> $Y_i = \text{FFN}_i(\dots \text{FFN}_2(\text{FFN}_1(X)) \dots),$
>
> only involving X, and no other tokens. Then we can simplify the transformation of multiple FFNs to a single 2-layer MLP, which results in SAISA. In contrast, when the transformation involves interactions with other tokens (as modeled by attention mechanism), it cannot be simplified to a simple MLP.
>
> Regarding the performance comparison, we find that SAISA achieves better performance than using NAAViT alone. We attribute this improvement to the following reason: when using NAAViT alone, we apply FFNs to both visual and text tokens. Since these FFN parameters in the LLM backbone are pre-trained on a large scale of text data, they are optimized for processing text tokens. Although we then train them on both visual and text tokens, as the two modalities demonstrate a clear modality gap in the embedding space [1], and the scale of optimized visual tokens is typically smaller than text tokens, these parameters cannot effectively handle both modalities. We believe this imbalance is also a reason of the "language-prior" phenomenon. In contrast, SAISA uses other MLPs to process visual tokens, and FFNs are optimized to process only text tokens.
>
> [1] Deciphering cross-modal alignment in large vision-language models with modality integration rate.

---

> ### Author Response · Authors · 2025-11-23
> **Author Response (2/2)**
>
> **[Weakness 3 & Question 2] Potential incompatibility of NAAViT masks with inference accelerators.**
>
> Thank you for raising this concern. We agree that NAAViT masks affect compatibility with inference accelerators, and the current implementation of NAAViT is based on scaled_dot_product_attention provided by PyTorch. However, as shown in Table 4, our SAISA still achieves **more than 1.65x speedup compared to LLaVA-1.5 with Flash-Attention 2**.
>
> For **multi-image and interleaved inputs**, we test SAISA on the multi-image subset of MMMU, which contains both multi-image and interleaved image-text input. SAISA achieves a 1.93x speed up, even higher than single-image inputs, because **as the number of visual tokens increases, SAISA achieves greater reduction in computational overhead**, as illustrated in Figure 7(b) of the revised paper. Moreover, NAAViT mask is deterministic and block-structured, meaning that **future engineering optimizations can make it fully compatible with both FlashAttention-2 and vLLM** inference engines.
>
> ***
> **[Weakness 4] Outdated baseline selection for token compression.**
>
> First, we compare our method with the tokens shuffle mechanism, which reduces token counts to 1/4. The results are shown below. SAISA **consistently outperforms the tokens shuffle method**.
>
> | Method | MMBench | MM-Vet | VQAv2 | GQA | POPE | VizWiz |
> |--------------|---------|--------|-------|------|------|--------|
> | Shuffle 1/4 | 64.0 | 29.7 | 76.2 | 60.1 | 85.9 | 48.8 |
> | SAISA | 65.7 | 30.5 | 77.7 | 60.9 | 87.2 | 52.3 |
>
> Then, regarding the comparison with parameter-free methods, we include newer baselines, such as TopV [1], Libra-Merging [2], VisionZip [3], SparseVLM [5], PAR [6], VisiPruner [7], expanding both token pruning and token merging. The results are shown below. We observe that SAISA **consistently outperforms these token compression methods**.
>
> | Method| TFLOPs | MMMU | MMBench-EN | MMBench-CN | POPE | ScienceQA | GQA |
> |-----------------------------|--------|-------|------------|------------|-------|-----------|-------|
> | TopV (CVPR 2025) | 4.35 | 35.1 | 59.65 | - | 84.21 | 69.61 | - |
> | Libra-Merging (CVPR 2025) | 3.16 | - | 63.9 | 58.2 | - | 69.2 | 60.7 |
> | VisionZip (CVPR 2025) | 2.84 | 36.2 | 64.3 | 56.7 | 84.9 | 68.2 | 60.1 |
> | SparseVLM (ICML 2025) | 2.84 | 33.8 | 60.0 | 51.1 | 80.5 | 67.1 | 59.5 |
> | PAR (ICCV 2025) | 4.00 | - | 60.2 | 50.7 | 73.6 | 69.6 | - |
> | VisiPruner (EMNLP 2025) | 3.93 | - | 62 | - | 84.4 | 66.7 | 60.3 |
> | SAISA | 2.86 | 36.9 | 65.7 | 59.0 | 87.2 | 70.1 | 60.9 |
>
> [1] TopV: Compatible Token Pruning with Inference Time Optimization for Fast and Low-Memory Multimodal Vision Language Model.
>
> [2] Libra-Merging: Importance-Redundancy and Pruning-Merging Trade-Off for Acceleration Plug-In in Large Vision-Language Model.
>
> [3] VisionZip: Longer is Better but Not Necessary in Vision Language Models.
>
> [4] SparseVLM: Visual Token Sparsification for Efficient Vision-Language Model Inference.
>
> [5] Pruning All-Rounder: Rethinking and Improving Inference Efficiency for Large Vision Language Models.
>
> [6] VisiPruner: Decoding Discontinuous Cross-Modal Dynamics for Efficient Multimodal LLMs.
>
> ***
> **[Weakness 5] Limited experimental scope.**
>
> We conduct experiments on the larger model with 13B parameters. We present the experimental results below. Our method **continues to perform well as the model size increases**.
>
> | Method | MMMU | MMBench-EN | MMBench-CN | POPE | ScienceQA | GQA |
> |----------------|-------|------------|------------|-------|-----------|------|
> | LLaVA-1.5-13B | 35.3 | 67.7 | 63.6 | 87.1 | 71.6 | 63.3 |
> | SAISA-13B | 35.9 | 68.1 | 64.1 | 87.3 | 71.7 | 62.9 |
>
> For **other modalities such as videos and multi-images inputs**, we apply our method to Mantis-clip, an MLLM for videos and multi-images inputs with fully open source training data. The experimental results are shown below. Our method still achieves superior performance on videos and multi-images scenarios.
>
> | Method | NLVR2 | Mantis-Eval | LongVideoBench | LVBench | NExTQA |
> |-----------------|-------|-------------|----------------|---------|--------|
> | Mantis| 84.7 | 53.0 | 45.1 | 35.5 | 72.7 |
> | Ours | 84.8 | 54.6 | 45.7 | 35.1 | 73.2 |

---

> > ### Comment · Reviewer_JnP5 · 2025-11-25
> >
> > Thank you for the substantial additional work provided during the rebuttal stage. The newly added redundancy analyses, along with the breakdown of FLOPs and performance contributions for NAAViT and SAISA, effectively alleviated my initial concerns regarding the soundness of the experimental setup and methodology. The inclusion of more baselines has also, to some extent, addressed my worries about the breadth of experimental coverage.
> >
> > Taking these additions and their increased persuasiveness into account, I am raising my score from 2 to 4. Although some limitations remain, the work now demonstrates practical value and meaningful reference significance overall.

---

### Official Review · Reviewer_rU2Y · 2025-10-30

**Soundness:** 3
**Presentation:** 3
**Contribution:** 3
**Rating:** 6
**Confidence:** 3

**Summary:**

This paper proposes SAISA (Self-Attention Input Space Alignment), a novel and highly efficient architecture for Multimodal Large Language Models (MLLMs). The work is motivated by a key finding: that self-attention among visual tokens within the LLM is "highly redundant." The authors first validate this by proposing NAAVIT (No Attention Among Visual Tokens), a modified self-attention mechanism that eliminates this $O(v^2)$ computation, and show it improves performance over standard attention.Building on this, the SAISA architecture aligns visual features directly with the input spaces of each NAAVIT self-attention block in the LLM. This clever design allows visual tokens to participate in attention (text can attend to visual tokens, visual tokens can attend to text) but bypasses two major computational bottlenecks: (1) self-attention among visual tokens and (2) all FFN computations for visual tokens.The paper demonstrates that, compared to the LLaVA-1.5 architecture, SAISA reduces inference FLOPs by 66% and the training budget by 26%, all while achieving superior performance on a wide range of MLLM benchmarks.

**Strengths:**

- A 66% reduction in inference FLOPs and a 26% reduction in training budget (vs. LLaVA-1.5) are extremely significant practical contributions. This makes SOTA-level MLLMs more accessible to train and cheaper to deploy.

- The core finding that attention among visual tokens is redundant—and that FFNs for visual tokens can also be skipped—is a fundamental and novel insight into MLLM architecture.

- This is not just an efficiency-focused paper (e.g., pruning, quantization). SAISA outperforms its LLaVA-1.5 baseline across a wide array of benchmarks (Tables 2, 3, 5), indicating the architectural changes also serve as a good regularizer or are simply a better design.

- The ablation studies in Table 5 compellingly show that SAISA's benefits are not specific to one model pair but apply broadly across different LLM families (including GQA models like Mistral) and visual encoder types (ViT, ConvNeXt).

**Weaknesses:**

- The paper does an excellent job proving that attention among visual tokens is redundant but doesn't fully explore why. Is it because the visual encoder has already "bound" all necessary spatial/object relationships? Does removing it act as a regularizer, preventing the model from "overthinking" the visual features? A deeper analytical probe (e.g., attention visualization, feature analysis) into why NAAVIT works better would make this paper even stronger.

- The SAISA projector consists of n separate 2-layer MLPs (one for each of the n LLM layers), whereas the LLaVA projector is a single 2-layer MLP. This significantly increases the number of new parameters. The paper shows that training is still 26% faster (which is the more important metric), likely due to reduced computation in the frozen LLM blocks. However, an explicit comparison of the number of new, trainable parameters (SAISA projector vs. LLaVA projector) would be helpful for transparency. The paper's "shared MLP" pre-training strategy seems to be the key to managing this, but this could be clearer.

**Questions:**

- The finding that NAAVIT (no visual self-attention) outperforms vanilla self-attention (Table 1) is fascinating. Do the authors have a hypothesis for why this is the case? Does it act as a regularizer? Or is it possible that the visual encoder's features are already so robust that further self-attention only adds noise?

- Could you please provide the concrete number of new, trainable parameters introduced by the SAISA projector (e.g., for Vicuna-7B with 32 layers) and compare this to the number of parameters in the LLaVA-1.5 projector? This would clarify the trade-off being made (more parameters in the projector for far less computation in the LLM).

- How does SAISA compare to orthogonal efficiency methods like visual token pruning (e.g., FastV, cited in the paper)? SAISA's approach is to keep all 576 tokens but process them more cheaply. Would it be better to (e.g.) prune to 64 tokens and use a standard LLaVA architecture? Or can these methods be combined (e.g., use SAISA with a resampler, as in Table 7)?

---

> ### Author Response · Authors · 2025-11-23
> **Author Response (1/n)**
>
> Thank you for providing your valuable and constructive feedback! Please find below the responses to each comment.
>
> ***
>
> **[Weakness 1 & Question 1] Reasons for why NAAViT outperforms vanilla self-attention.**
>
> Thank you for the insightful comment. We have added a new subsection, **Subsection 3.2 Redundancy Analyses**, in the revised paper. This new subsection introduces several complementary analyses that strengthen our evidence:
>
> 1. **Cosine similarity-based token feature analysis.** We measure the transformation strength that each self-attention block applies to visual tokens by computing the cosine similarities between the input and output tokens at each block. The results show that self-attention block in shallow layers apply almost no transformation to visual tokens, indicating strong redundancy at the early stages of the model.
>
> 2. **Attention map visualizations.** We visualize attention heatmaps. In deeper layers—those directly influencing the final prediction—visual tokens receive extremely little attention from model's output tokens. This suggests that even if some transformations occur on visual tokens in deeper layers, they have almost no downstream influence.
>
> 3. **KL divergence–based ablations.** To quantify contribution more directly, we remove each attention block’s transformations on visual tokens (while keeping text-token transformations intact) and compute the KL divergence between the modified model’s output logits and those of the original model. Across all layers, ablating transformations on visual tokens yields consistently lower KL divergence than ablating those on text tokens, showing that visual-token updates have minimal impact on the model’s output.
> Together, these analyses provide more direct and quantitative support for our claim, thereby reinforcing the motivation behind NAAViT.
>
> We also clarify why such redundancy arises. First, MLLMs **apply causal masking to visual tokens, even though images are not sequential**. This artificial ordering **introduces unnecessary noise**. Second, **modern visual encoders already capture rich relationships among patches**, making additional attention among visual tokens inside the LLM largely redundant, confirmed by the small KL divergences when these transformations are removed.

---

> ### Author Response · Authors · 2025-11-23
> **Author Response (2/n)**
>
> **[Weakness 2 & Question 2] Comparison of new, trainable parameters**
>
> Thank you for your question and apologize for any unclear explanation. The number of new parameters introduced by the SAISA projector is **671M**, while the number of parameters in the LLaVA-1.5 projector is 21M. **Compared to the LLM, the SAISA projector only introduces less than 10% parameters**.
>
> During pre-training, we use a single 2-layer MLP to project visual feature into the LLM hidden state space, and these projected visual tokens are directly injected into different LLM layers. This means that **only 21M parameters are trainable during pre-training, same as LLaVA-1.5**. However, **SAISA achieves more than 100% faster pre-training than LLaVA-1.5**, because it significant reduces computational complexity in the LLM.
>
> For fine-tuning, we initialize the n separate MLPs from the pre-trained MLP, and the total trainable parameters include the 7B in the LLM and the 671M in the projector. **Although SAISA have more trainable parameters than LLaVA-1.5 during fine-tuning, it's training is still 21% faster than LLaVA-1.5 in this stage**, because it substantially reduces computation in the LLM.
>
> **Discussion:** Although SAISA introduces ~9.6% new parameters to the LLM, it achieves **improvement in both efficiency and performance compared to LLaVA-1.5**. The difference between the SAISA projector and the LLaVA-1.5 projector is that the SAISA projector directly projects visual features into each LLM layer, whereas the LLaVA-1.5 projector projects visual features into the LLM embedding space, and relies on FFNs in the LLM to transform them across layers. We compare SAISA with the model using NAAViT alone (in Table 1) to make a comparison between the SAISA projector and the LLaVA-1.5 projector, as these models both use NAAViT. We list the comparison as follows.
>
> | Method | TFLOPs | MMMU | MMBench-EN | MMBench-CN | POPE | ScienceQA | OKVQA |
> |-----------|--------|------|------------|------------|------|-----------|-------|
> | NAAViT (LLaVA-1.5 projector) | 7.10 | 36.0 | 64.9 | 58.4 | 86.9 | 68.7 | 56.0 |
> | SAISA (SAISA projector) | 2.86 | 36.9 | 65.7 | 59.0 | 87.2 | 70.1 | 60.9 |
>
> We find that the SAISA projector achieves better performance than the LLaVA-1.5 projector. We attribute this improvement to the following reason: when using NAAViT alone, we apply FFNs to both visual and text tokens. Since these FFN parameters in the LLM are pre-trained on a large scale of text data, they are optimized for processing text tokens. Although we then train them on both visual and text tokens, as the two modalities demonstrate a clear modality gap in the embedding space [1], and the scale of optimized visual tokens is typically smaller than text tokens, these parameters cannot effectively handle both modalities. We believe this imbalance is also a reason of the "language-prior" phenomenon. In contrast, SAISA uses other MLPs in its projector to process visual tokens, and FFNs are optimized to process only text tokens.
>
> [1] Deciphering cross-modal alignment in large vision-language models with modality integration rate.

---

> ### Author Response · Authors · 2025-11-23
> **Author Response (3/3)**
>
> **[Question 3] Comparison between SAISA and orthogonal efficiency methods.**
>
> Thank you for your question! **We kindly note that we have already provided a direct comparison between SAISA and visual token pruning, including FastV and VTW, in Table 2 of our original paper**. To make more comprehensive comparison, we include **many newer orthogonal efficiency methods**, such as TopV (CVPR 2025) [2], Libra-Merging (CVPR 2025) [3], VisionZip (CVPR 2025) [4] , SparseVLM (ICML 2025) [5], PAR (ICCV 2025) [6], VisiPruner (EMNLP 2025) [7], expanding both token pruning and token merging. The results are as follows.
>
> | Method | TFLOPs | MMMU | MMBench-EN | MMBench-CN | POPE | ScienceQA | GQA |
> |--------------|--------|------|------------|------------|-------|-----------|------|
> | FastV | 4.90 | 35.8 | 64.3 | 56.8 | 82.45 | 67.8 | 60.2 |
> | VTW | 4.70 | 36.3 | 64.0 | 58.5 | 86.0 | 69.1 | 55.1 |
> | TopV | 4.35 | 35.1 | 59.7 | - | 84.2 | 69.6 | - |
> | Libra-Merging| 3.16 | - | 63.9 | 58.2 | - | 69.2 | 60.7 |
> | VisionZip | 2.84 | 36.2 | 64.3 | 56.7 | 84.9 | 68.2 | 60.1 |
> | SparseVLM | 2.84 | 33.8 | 60.0 | 51.1 | 80.5 | 67.1 | 59.5 |
> | PAR | 4.00 | - | 60.2 | 50.7 | 73.6 | 69.6 | - |
> | VisiPruner | 3.93 | - | 62.0 | - | 84.4 | 66.7 | 60.3 |
> | SAISA | 2.86 | 36.9 | 65.7 | 59.0 | 87.2 | 70.1 | 60.9 |
>
> We observe that **SAISA consistently outperforms these methods**. Moreover, **our method and most token pruning methods can be combined**. For example, we apply VTW to SAISA, and the experimental results are as follows. Combining both methods further improves MLLM efficiency, and SAISA + VTW outperforms LLaVA-1.5 + VTW, **demonstrating strong compatibility between SAISA and token pruning**.
>
> | Method | TFLOPs | MMMU | MMBench-EN | MMBench-CN | POPE | ScienceQA | GQA |
> |------------------|--------|------|------------|------------|------|-----------|------|
> | LLaVA-1.5 + VTW | 4.70 | 36.3 | 64.0 | 58.5 | 86.0 | 69.1 | 55.1 |
> | SAISA | 2.86 | 36.9 | 65.7 | 59.0 | 87.2 | 70.1 | 60.9 |
> | SAISA + VTW | 1.85 | 36.7 | 65.6 | 58.8 | 87.0 | 70.2 | 57.7 |
>
> [1] Eagle: Exploring The Design Space for Multimodal LLMs with Mixture of Encoders.
>
> [2] TopV: Compatible Token Pruning with Inference Time Optimization for Fast and Low-Memory Multimodal Vision Language Model.
>
> [3] Libra-Merging: Importance-Redundancy and Pruning-Merging Trade-Off for Acceleration Plug-In in Large Vision-Language Model.
>
> [4] VisionZip: Longer is Better but Not Necessary in Vision Language Models.
>
> [5] SparseVLM: Visual Token Sparsification for Efficient Vision-Language Model Inference.
>
> [6] Pruning All-Rounder: Rethinking and Improving Inference Efficiency for Large Vision Language Models.
>
> [7] VisiPruner: Decoding Discontinuous Cross-Modal Dynamics for Efficient Multimodal LLMs.

---

### Author Response · Authors · 2025-12-03
**General Response (1/2)**

Dear AC,

We would like to thank you for your hard work during this exceptional period. Considering your additional work this time, we provide a summary of the rebuttal as follows.

We sincerely thank all reviewers for their constructive feedback and comments. We have addressed every concern with comprehensive supplementary experiments and detailed explanation.

During the rebuttal, **Reviewer JnP5** confirmed that our responses **"effectively alleviated my initial concerns regarding the soundness of the experimental setup and methodology"**, and our work **"demonstrates practical value and meaningful reference significance"**. He/She **raised the score from 2 to 4**.

**Reviewers rU2Y (Rating 6), Cqp8 (Rating 4) and QKS9 (Rating 4) have not had the chance to respond to our rebuttal**, but we provide substantial additional experiments and clarification as requested. We are confident that these responses can lead to further score increases if the rebuttal process as initial planned.

We summarize the reviewers' concerns and our responses below.

***

## **Reviewer rU2Y (Rating 6, Soundness 3, Presentation 3, Contribution 3)**

**[W1, Q1] Why NAAViT outperforms vanilla attention (also related to Reviewer JnP5 W1)**
We provide analyses in Section 3.2, including attention visualization and feature analysis, providing evidence for the effectiveness of NAAViT. We attribute this to causal mask on visual tokens and robustness of visual encoders.

**[W2, Q2] Comparison of new, trainable parameters**
We list detailed comparison of new, trainable parameters and training speed, clarify the pre-training strategy, and provide motivation for such projector design.

**[Q3] Comparison with orthogonal efficiency methods**
**We emphasize that we have already provided comparison with such methods, eg. FastV, in Table 2 of our original paper.** We also compare to many newer orthogonal efficiency methods. **SAISA outperforms all these methods.** Moreover, we conduct experiments showing **compatibility between SAISA and these methods**.

***

## **Reviewer JnP5 (Rating 2 -> 4)**

**[W1] More evidence for redundancy**
Similar to the response for **Reviewer rU2Y W1, Q1**, we provide analyses in Subsection 3.2, including attention heatmaps and token correlation analyses. We also provide tests on multi-object and spatial reasoning tasks, demonstrating visual–visual interactions are dispensable.

**[W2, Q1] Unclear source of efficiency gains, and results of NAAViT**
We present a direct comparison between NAAViT and SAISA. Both NAAViT and removing FFNs on visual tokens contribute to acceleration.

**[W3, Q2] Potential incompatibility with inference engines**
We agree that NAAViT affect compatibility with inference accelerators **currently**, but SAISA is still **1.65x faster than the baseline with Flash-Attention 2**. For multi-image and interleaved inputs, SAISA achieves even higher speedup. Moreover, **future engineering optimizations can make it fully compatible with these engines**.

**[W4] Newer token compression baselines**
We compare with token shuffle and many newer baselines in CVPR25, ICML25, ICCV25 and EMNLP25. **SAISA outperforms all these baselines**.

**[W5] Experiments on 13B model/video/multi-image input**
We conduct experiments on the 13B model and Mantis (for video & multi-image input). Our method outperforms baselines in these scenarios.

***

## **Reviewer Cqp8 (Rating 4, Soundness 3, Presentation 3, Contribution 3)**

**[W1] Novelty**
We clarified the novelty and motivation behind our design. Moreover, prior work improves one side of training/inference efficiency but not both. **SAISA achieves both, as well as superior performance**.

**[W2] Comparison between NAAViT and SAISA, and more ablation (related to Reviewer JnP5 W2, Q1)**
We present a direct comparison between NAAViT and SAISA. **Both designs (NAAViT, layer-wise projection) improve performance and efficiency, and the former is the foundation of the latter**. We conduct ablation on subsets of layers (early, middle, late, all). **Projecting to all layers gives the best performance. Early layers are better than middle/late layers.**

**[W3] Experiments on newer models and comparison with orthogonal efficiency methods (related to Reviewer rU2Y Q3)**
As the reviewer requested comparison on more recent architecture that have been published in late 2024 or 2025, we integrate our method to **EAGLE (ICLR25), due to its open source training data**. Our method outperforms EAGLE. **We emphasize that we have already provided comparison with orthogonal efficiency methods, eg. FastV, in Table 2 of our original paper**, as mentioned in the response to Reviewer rU2Y Q3.

---

### Author Response · Authors · 2025-12-03
**General Response (2/2)**

## **Reviewer QKS9 (Rating 4, Soundness 3, Presentation 3, Contribution 2)**

**[W1] Comparison with prior work [1]**
We dicuss differences between our work and [1], as well as other prior work, in Subsection 6.2.
[1] Implicit multimodal alignment: On the generalization of frozen llms to multimodal inputs. Mustafa Shukor et al

**[W2, W3]  Experiments on stronger baseline, scaled data and scaled model (related to Reviewer JnP5 W4, W5)**
Since the training data of most SOTA MLLMs is not opensource, we integrate our method to **EAGLE, a stronger baseline with scaled data**. Our method outperforms EAGLE. We scale model size to **13B**. SAISA still outperforms the baseline.

**[W4] Experiments on high-resolution and fine-grained benchmarks**
We conduct evaluations on multiple such benchmarks. Our method achieves **similar performance with the baseline, while significantly improving efficiency**.

---

### Meta-Review · Area_Chair_LpL4 · 2026-01-07

**Summary:**

This paper investigates the computational redundancy of self-attention among visual tokens in Multimodal Large Language Models (MLLMs). It proposes **SAISA**, an architecture that eliminates visual-visual attention (NAAVIT) and skips Feed-Forward Network (FFN) operations for visual tokens by aligning features directly to LLM input spaces. The authors report a 66% reduction in inference FLOPs and a 26% reduction in training costs compared to LLaVA-1.5.

### Strengths

- **Efficiency Gains**: The architecture provides significant practical improvements in both training and inference speed.

- **Empirical Validation**: The findings are supported by a wide range of benchmarks and various LLM backbones.

- **Detailed Rebuttal**: The authors provided extensive new analyses during the rebuttal, including KL-divergence and scaling tests to 13B models.

### Weaknesses

- **Incremental Novelty**: The core ideas are viewed as clever engineering combinations (masking and layer-wise projection) rather than a fundamental architectural shift.

- **Projector Complexity**: The SAISA projector significantly increases the number of trainable parameters (671M vs. 21M in LLaVA-1.5), adding overhead during fine-tuning.

- **Inference Compatibility**: The specialized attention masks may face compatibility issues with standard inference accelerators like vLLM or Flash-Attention 2.


### Reason for Rejection

Despite the impressive efficiency metrics, the decision to reject is based on the following:

- **Concerns over Soundness**: Even after the rebuttal, a key reviewer (JnP5) only moved to a "Marginal Below" score (4), indicating lingering doubts about the work's foundational impact.

- **Marginal Contribution**: Multiple reviewers (Cqp8, QKS9) rated the contribution as "fair" or "marginal," suggesting that the conceptual leap does not meet the high bar for ICLR.

- **Architectural Trade-offs**: The significant increase in projector parameters and potential friction with optimized inference engines limit the generalizability and elegance of the proposed solution.

**Reviewer Concerns:**

Based on the author responses and the final reviewer scores, the following concerns were addressed or remain:

### Addressed Concerns

- **Insufficient Evidence for Redundancy:** The authors provided quantitative evidence through KL-divergence, cosine similarity-based feature analysis, and attention heatmaps, proving visual-token updates have minimal impact on output.

- **Disentangling Efficiency Gains:** The rebuttal explicitly broke down the reduction in FLOPs, attributing 17% to NAAVIT and 49% to the removal of FFNs.

- **Outdated Baselines:** The authors integrated their method into the stronger **EAGLE (ICLR 2025)** baseline and compared SAISA against multiple 2025 efficiency methods like TopV, VisionZip, and SparseVLM.



- **Scaling and Generalization:** New experiments on 13B models , video/multi-image inputs (Mantis) , and fine-grained benchmarks (TextVQA, OCRBench) were provided.

- **Architecture Ablations:** The authors performed layer-wise ablation studies, showing that projecting features to all layers is optimal and identifying that early layers are most critical for integration.



### Outstanding Concerns

- **Incremental Novelty:** Despite the authors' clarifications, Reviewer **Cqp8** still viewed the mechanism as a "clever engineering combination" rather than a foundational conceptual leap.



- **Inference Accelerator Compatibility:** While SAISA shows empirical speedups (1.65x with Flash-Attention 2), the specific "fragmented" attention masks in NAAVIT may still require future engineering optimizations to be fully native to all inference engines like vLLM.


- **Parameter Trade-off:** The SAISA projector introduces significantly more trainable parameters during fine-tuning (671M) compared to the LLaVA-1.5 projector (21M), a trade-off for the reduced computation in the LLM backbone.

**Reviewer Scores:**

**Reviewer JnP5 (Initial Rating: 2)**

**Confirmed Score: 2$\to$4**

**Reasoning**: This reviewer officially raised their score after the rebuttal, stating that the authors' "substantial additional work" and "newly added redundancy analyses" effectively alleviated their initial concerns regarding methodology and soundness. The reviewer noted that the work now demonstrates "practical value and meaningful reference significance".


**Other Reviewers (rU2Y, Cqp8, QKS9)**


**Status**: As of the final author summary, these reviewers had **not** yet responded to the rebuttal to update their scores. Their last recorded ratings remain **6, 4, and 4** respectively.

---

### Decision · Program_Chairs · 2026-01-26

Reject